# Melanopsin activates divergent phototransduction pathways in intrinsically photosensitive retinal ganglion cell subtypes

Ely Contreras[1,2], Jacob D Bhoi[1,3], Takuma Sonoda[1,3], Lutz Birnbaumer[4,5], Tiffany M Schmidt[1,6]*

[1]Department of Neurobiology, Northwestern University, Evanston, United States; [2]Northwestern University Interdisciplinary Biological Sciences Program, Northwestern University, Evanston, United States; [3]Northwestern University Interdepartmental Neuroscience Program, Northwestern University, Chicago, United States; [4]Laboratory of Signal Transduction, National Institute of Environmental Health Sciences, Durham, United States; [5]Institute of Biomedical Research (BIOMED), Catholic University of Argentina, Buenos Aires, Argentina; [6]Department of Ophthalmology, Feinberg School of Medicine, Chicago, United States

*For correspondence:
tiffany.schmidt@northwestern.
edu

Competing interest: The authors declare that no competing interests exist.

**Abstract** Melanopsin signaling within intrinsically photosensitive retinal ganglion cell (ipRGC) subtypes impacts a broad range of behaviors from circadian photoentrainment to conscious visual perception. Yet, how melanopsin phototransduction within M1-M6 ipRGC subtypes impacts cellular signaling to drive diverse behaviors is still largely unresolved. The identity of the phototransduction channels in each subtype is key to understanding this central question but has remained controversial. In this study, we resolve two opposing models of M4 phototransduction, demonstrating that hyperpolarization-activated cyclic nucleotide-gated (HCN) channels are dispensable for this process and providing support for a pathway involving melanopsin-dependent potassium channel closure and canonical transient receptor potential (TRPC) channel opening. Surprisingly, we find that HCN channels are likewise dispensable for M2 phototransduction, contradicting the current model. We instead show that M2 phototransduction requires TRPC channels in conjunction with T-type voltage-gated calcium channels, identifying a novel melanopsin phototransduction target. Collectively, this work resolves key discrepancies in our understanding of ipRGC phototransduction pathways in multiple subtypes and adds to mounting evidence that ipRGC subtypes employ diverse phototransduction cascades to fine-tune cellular responses for downstream behaviors.

## Editor's evaluation

Retinal ganglion cells with intrinsic photosensitivity play important emerging physiological roles. The mechanisms of phototransduction are still not well known and there exists controversy regarding the ion channels responsible for the photo response. The authors of this article present convincing data that contribute to understanding the ionic mechanisms in two of these cell types. This article will be of general interest to biologists and neuroscientists and should help resolve a major issue in retinal physiology.

## Introduction

Light is a pervasive and important regulator of physiology and behavior across timescales that range from milliseconds to days. While rod and cone photoreceptors are primarily responsible for rapid, spatially discrete light signals, melanopsin phototransduction within the M1-M6 intrinsically photo-sensitive retinal ganglion cell (ipRGC) subtypes spatially and temporally integrates environmental light signals to impact diverse behaviors over significantly longer timescales (*Berson et al., 2002*; *Hattar et al., 2002*; *Wong et al., 2005*; *Wong, 2012*; *Emanuel and Do, 2015*). M1 ipRGCs project to non-image-forming brain regions to influence subconscious non-image-forming functions such as circadian photoentrainment, the pupillary light reflex, learning, and mood (*Hattar et al., 2003*; *Lucas et al., 2003*; *Mrosovsky and Hattar, 2003*; *Panda et al., 2003*; *Altimus et al., 2008*; *Lupi et al., 2008*; *Göz et al., 2008*; *Gooley et al., 2012*; *LeGates et al., 2012*; *Fernandez et al., 2018*; *Rupp et al., 2019*; *Sondereker et al., 2020*; *Aranda and Schmidt, 2021*). M2-M6 ipRGCs primarily innervate brain areas involved in conscious visual perception and are necessary for proper contrast sensitivity (*Ecker et al., 2010*; *Estevez et al., 2012*; *Zhang et al., 2023*; *Schmidt et al., 2014*; *Stabio et al., 2018*; *Quattrochi et al., 2019*; *Aranda and Schmidt, 2021*). Despite diverse behavioral functions, one common feature across ipRGC subtypes is that melanopsin (Opn4) phototransduction is required for intrinsic light sensitivity. Indeed, melanopsin null mutant animals show deficits in both image-forming and non-image-forming behaviors (*Panda et al., 2002*; *Ruby et al., 2002*; *Lucas et al., 2003*; *Schmidt et al., 2014*), highlighting the important role melanopsin plays across diverse cell types and behaviors. Though all ipRGC subtypes require melanopsin phototransduction for intrinsic photosensitivity, the melanopsin-mediated response of each subtype differs in size, sensitivity, and kinetics and will differentially shape the visual information relayed to the brain (*Graham et al., 2008*; *Schmidt and Kofuji, 2009*; *Ecker et al., 2010*; *Perez-Leighton et al., 2011*; *Estevez et al., 2012*; *Zhang et al., 2023*; *Jiang et al., 2018*; *Sonoda et al., 2018*; *Stabio et al., 2018*; *Quattrochi et al., 2019*). Yet, the mechanistic underpinnings of these key differences and how they impact visual signals relayed by ipRGC subtypes to drive diverse downstream behaviors are not well understood.

Melanopsin shares greater sequence homology with invertebrate rhodopsins than vertebrate opsins, which led to an early expectation that melanopsin phototransduction in all ipRGC subtypes would work through an identical, invertebrate-like signaling pathway involving activation of a Gq/PLC-based cascade that opens canonical transient receptor potential (TRPC) channels such as TRPC 3, 6, or 7 (*Provencio et al., 1998*; *Provencio et al., 2000*; *Koyanagi et al., 2005*; *Warren et al., 2006*; *Hartwick et al., 2007*; *Graham et al., 2008*; *Koyanagi and Terakita, 2008*; *Perez-Leighton et al., 2011*; *Graham et al., 2008*; *Xue et al., 2011*). However, though M1 ipRGCs use this predicted Gq/PLC-based cascade to open only TRPC6/7 channels, phototransduction in non-M1 ipRGCs has been reported to rely less exclusively on TRPC channels and to also target additional channel types (*Warren et al., 2006*; *Hartwick et al., 2007*; *Graham et al., 2008*; *Xue et al., 2011*; *Perez-Leighton et al., 2011*; *Sonoda et al., 2018*; *Jiang et al., 2018*; *Contreras et al., 2021*). In M2 cells, for example, melanopsin phototransduction is reported to open both TRPC channels and hyperpolarization-activated cyclic nucleotide-gated (HCN) channels (*Perez-Leighton et al., 2011*; *Jiang et al., 2018*; *Contreras et al., 2021*). In M4 cells, researchers have reached conflicting conclusions about the identity of the melanopsin transduction channel(s). Our previous findings in M4 cells point to closure of potassium leak channels by melanopsin phototransduction and a minor contribution from TRPC3/6/7 channel opening, while a concurrent study concluded that melanopsin phototransduction leads to opening of HCN channels in M4 cells with no contribution from TRPC3/6/7 channels (*Sonoda et al., 2018*; *Jiang et al., 2018*; reviewed in *Contreras et al., 2021*). Potassium channel closure versus HCN channel opening would lead to distinct, largely opposing impacts on M4 cell physiology and signaling, and thus a major goal of this study was to resolve the role of HCN and TRPC channels in M4 phototransduction, as well as to reassess their role in M2 phototransduction.

In this study, we show that HCN channels are not the M4 phototransduction channel and confirm a minor role for TRPC3/6/7 channels in bright light. Unexpectedly, we also find that HCN channels are dispensable for M2 phototransduction and identify T-type voltage-gated calcium channels as a novel, and critical, component of M2 phototransduction. Unlike M2 cells, M1 cells do not require T-type voltage-gated calcium channels for phototransduction, highlighting an important difference between the TRPC-dependent phototransduction cascades of M1 versus M2 ipRGCs. Thus, M1, M2, and M4 phototransduction cascades each signal through distinct combinations of phototransduction channels.

Collectively, our findings resolve important discrepancies in our understanding of ipRGC phototransduction and add to the growing body of evidence for diverse melanopsin signaling cascades across ipRGC subtypes that tune cellular function to drive diverse downstream behaviors.

## Results
### TRPC3/6/7 channels contribute to M4 phototransduction

Two models for M4 phototransduction have been proposed. Our previous work suggests that melanopsin in M4 ipRGCs leads to potassium channel closure with a minor contribution of TRPC3/6/7 channel

**Table 1.** Side-by-side comparison of parameters and cell subtype identification criteria in the current study and *Jiang et al., 2018*. Citations provided in 'Materials and methods'.

| | Current study | | Jiang et al., 2018 | |
|---|---|---|---|---|
| | **M2** | **M4** | **M2** | **M4** |
| Mouse models | Opn4-GFP *Trpc3⁻/⁻*; *Trpc6⁻/⁻*; *Trpc7⁻/⁻* | WT *Trpc3⁻/⁻*; *Trpc6⁻/⁻*; *Trpc7⁻/⁻* | Opn4-tdTomato Primarily: *Trpc6⁻/⁻*; *Trpc7⁻/⁻* *Figure 2*: *Trpc1⁻/⁻*; *Trpc3⁻/⁻*; *Trpc4⁻/⁻*; *Trpc5⁻/⁻*; *Trpc6⁻/⁻*; *Trpc7⁻/⁻* | Opn4-tdTomato Primarily: *Trpc6⁻/⁻*; *Trpc7⁻/⁻* *Figure 2*: *Trpc1⁻/⁻*; *Trpc3⁻/⁻*; *Trpc4⁻/⁻*; *Trpc5⁻/⁻*; *Trpc6⁻/⁻*; *Trpc7⁻/⁻* |
| Cell targeting ex vivo | Epifluorescence intensity | IR-DIC/soma size and ON-sustained light response | Epifluorescence intensity/soma size | Epifluorescence intensity/soma size |
| Subtype identity | During recording:<br>• Stratification analysis using Alexa 594<br>• Neurobiotin fill for post-hoc morphological analysis<br><br>Post recording:<br>• SMI-32 negative<br>• ON-stratification<br>• Large arbors with moderate branching<br><br>Defining feature: M2 ipRGCs are the only ON stratifying ipRGC labeled in Opn4-GFP mice and they are SMI-32 negative. | During recording:<br>• Stratification analysis using Alexa 594<br>• Neurobiotin fill for post hoc morphological analysis<br><br>Post recording:<br>• SMI-32 positive<br>• ON stratification<br>• Highly branched arbors<br><br>Defining feature: M4 ipRGCs are the only SMI-32 positive ipRGC, and are ON stratifying | Intracellular dye filling: Alexa 568 for morphological analysis of dendritic arbors (criteria unspecified) | Intracellular dye filling: Alexa 568 for morphological analysis of dendritic arbors (criteria unspecified) |
| Mice dark-adapted | Overnight | | 3 hr | |
| Technique | Whole-cell voltage-clamp recording | | Whole-cell voltage-clamp recording | |
| Holding potential | –66 mV for all experiments unless otherwise mentioned | | –66 mV for all experiments except *Figure 4* | |
| Internal Solution (mM) | 120 K-gluconate, 5 NaCl, 4 KCl, 10 HEPES, 2 EGTA, 4 ATP-Mg, 0.3 GTP-Na$_2$ and 7-Phosphocreatine-Tris, with the pH adjusted to 7.3 with KOH | | 120 K-gluconate, 5 NaCl, 4 KCl, 10 HEPES, 2 EGTA, 4 ATP-Mg, 0.3 GTP-Na$_2$ and 7-Phosphocreatine-Tris, with the pH adjusted to 7.3 with KOH | |
| Synaptic Blockers | 100 µM DNQX, 20 µM L-AP4, 100 µM picrotoxin, and 20 µM strychnine | | 20 µM DNQX, 50 µM AP5, 100 µM Hexamethonium, 100 µM picrotoxin, and 1 µM Strychnine | |
| Recording Temperature | 30–32°C | | 30–32°C | |
| ZD7288 Conditions | Concentration: 50 µM Incubation time for effective HCN blockade: 5–8 min Incubation time driving off-target effects: 20 min | | Concentration: 50 µM Incubation time: not reported | |
| Light step | 50 ms | | 200 ms | |
| Light intensity | $6.08 \times 10^{15}$ photons · cm$^{-2}$ · s$^{-1}$ blue LED light (480 nm) | | White light of an intensity equivalent to $1.75 \times 10^{18}$ photons cm$^{-2}$ s$^{-1}$ of 480 nm light for melanopsin (conversion done by response-matching in the linear range) | |

IR-DIC, infrared differential interference contrast; ipRGCs, intrinsically photosensitive retinal ganglion cells.

opening in bright light (*Sonoda et al., 2018*), while a separate study published concurrently proposed that HCN channels are opened by the melanopsin phototransduction pathway with no contribution from TRPC channels (*Jiang et al., 2018*). To begin to resolve these discrepancies, we first recorded the M4 photocurrent under conditions that matched those used in Jiang et al. as closely as possible (*Table 1* and see 'Materials and methods'). We stimulated M4 cells with brief, full-field, 50 ms flashes of high photopic ($6.08 \times 10^{15}$ photons · cm$^{-2}$ · s$^{-1}$) 480 nm light, the highest possible intensity for our LED light source. We identified large somata of putative M4 cells under infrared differential interference contrast (IR-DIC) and filled each recorded cell with Neurobiotin. We then confirmed the identity of each M4 cell post-recording using multiple, established criteria including morphology (verified ON stratification, large somata, highly branched dendritic arbors), physiology (ON-sustained responses to increments in light), and immunolabeling for SMI-32 (*Schmidt et al., 2014*; *Lee and Schmidt, 2018*; *Sonoda et al., 2018*; *Sonoda et al., 2020*; *Figure 1A*). We recorded the M4 melanopsin photocurrent in a cocktail of synaptic blockers (see 'Materials and methods') at –66 mV following exposure to a brief, 50 ms full-field flash. Control M4 cells consistently exhibited an inward photocurrent that was composed of both a relatively fast, transient component followed by a slower, larger inward current that persisted for more than 30 s after termination of the 50 ms light pulse (*Figure 1B and C*). This small, transient component was previously noted in a subset of M2 cells but has not been observed in M4 cells in photopic light, potentially due to the previous use of lower intensity photopic stimuli (*Sonoda et al., 2018*) or the use of bright, epifluorescent illumination to localize M4 cells that were subsequently recorded in high photopic light (*Jiang et al., 2018*; *Ecker et al., 2010*).

We next assessed the role of TRPC3/6/7 channels in M4 phototransduction. Our previous work described a relatively minor, but detectable, role for TRPC3/6/7 channels in M4 phototransduction in photopic light ($10^{12}$ photons · cm$^{-2}$ · s$^{-1}$) (*Sonoda et al., 2018*), while others have found no contribution of these channels at higher photopic light intensities ($10^{18}$ photons · cm$^{-2}$ · s$^{-1}$) (*Jiang et al., 2018*). To determine the role of TRPC3/6/7 channels in M4 phototransduction, we compared the intrinsic photocurrent of control M4 cells to that of M4 cells in *Trpc3$^{-/-}$; Trpc6$^{-/-}$; Trpc7$^{-/-}$* (TRPC3/6/7 KO) retinas to 50 ms full-field flashes of the same bright, full-field, 480 nm light ($6.08 \times 10^{15}$ photons · cm$^{-2}$ · s$^{-1}$) (*Figure 1C and D*). We analyzed the photocurrent amplitude at Early, Intermediate, and Late timepoints in the recording, as well as the maximum photocurrent amplitude (*Figure 1D* and see 'Materials and methods'). We found a significant decrease in the amplitude of the Early, transient, component in TRPC3/6/7 KO M4 cells, suggesting that this component is largely driven by TRPC3/6/7 channels. This was not due to changes in current density across genotypes because capacitance was not significantly different between control and TRPC3/6/7 M4 cells (*Figure 1—figure supplement 1*). Input resistance was also similar across genotypes (*Figure 1—figure supplement 1*). We also observed a slight, but not significant, decrease in all other quantified amplitudes. Collectively, these data are consistent with our previous findings that TRPC3/6/7 channels make a minor contribution to the M4 light response in bright light (*Figure 1C and D*; *Sonoda et al., 2018*).

## HCN channels are not required for M4 phototransduction

We next examined the reported contribution of HCN channels to M4 phototransduction. HCN channels are a class of nonspecific cation channels opened primarily by membrane hyperpolarization whose activation voltage shifts in the presence of cyclic nucleotides (*Biel et al., 2009*). One compelling piece of evidence in support of melanopsin phototransduction opening HCN channels was a reduction of the M4 photocurrent following application of 50 µM ZD7288, an HCN antagonist (*Jiang et al., 2018*). We therefore first tested whether 50 µM ZD7288 blocked both the M4 photocurrent and HCN-mediated tail current using similar recording conditions to Jiang et al., combined with rigorous post hoc subtype identification of M4 cells. Because the incubation period for ZD7288 was not reported by Jiang et al., we first tested whether a period of 5–8 min incubation with ZD7288 was sufficient to abolish the HCN current in M4 ipRGCs. To do this, we recorded HCN tail currents from M4 ipRGCs first in the absence and then in the presence of 50 µM ZD7288 (*Figure 2A*; *Van Hook and Berson, 2010*; *Chen and Yang, 2007*). We found that treatment with 50 µM ZD7288 for 5–8 min was sufficient to eliminate the HCN tail current and achieve full HCN channel blockade in M4 ipRGCs (*Figure 2A*). After confirming tail current blockade by 5–8 min of 50 µM ZD7288, we then measured the photocurrent of these same M4 cells where HCN channels are fully blocked. If HCN channels are the sole target of melanopsin phototransduction in M4 cells, then this full blockade of HCN channels with 5–8 min of

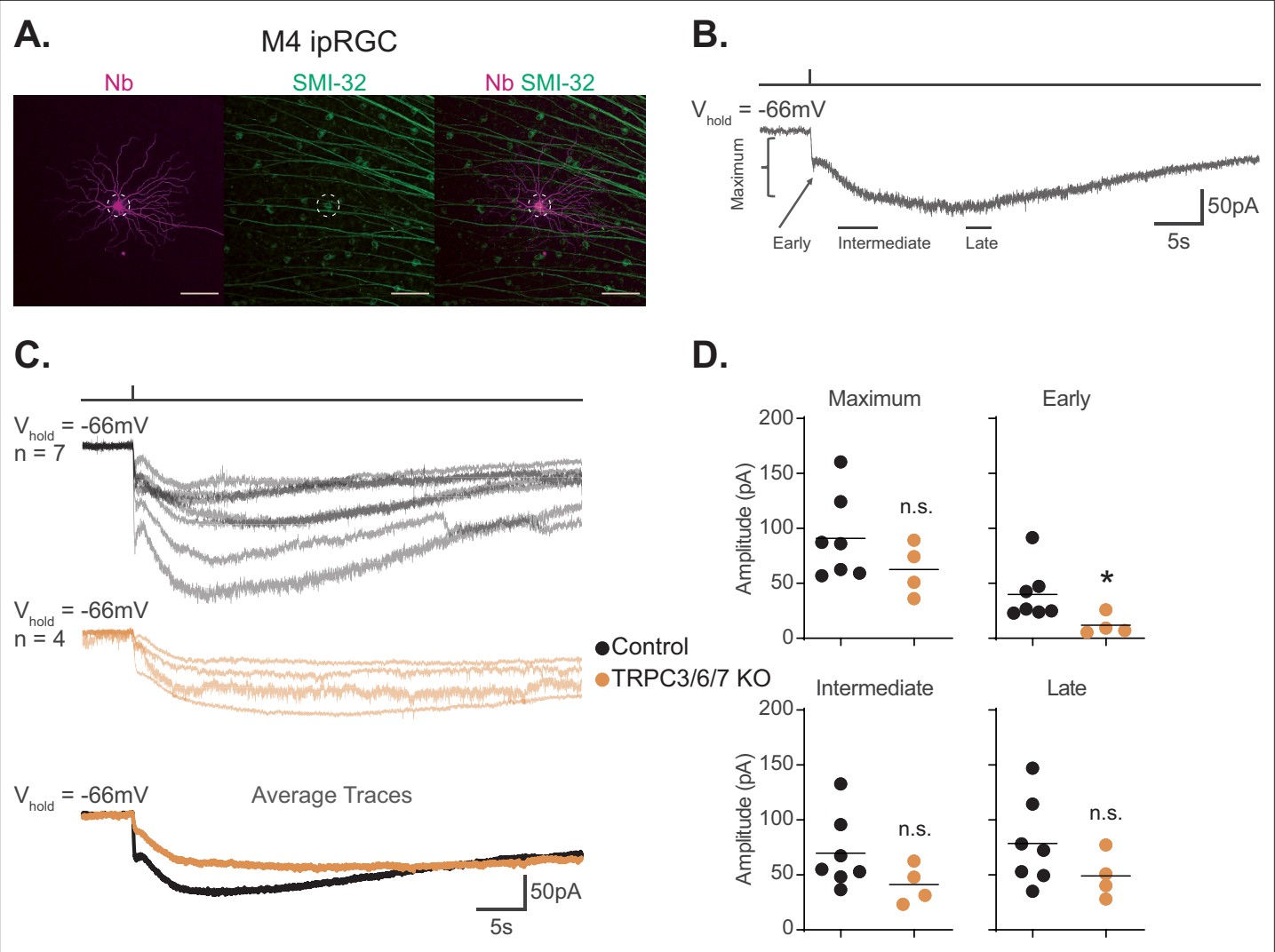

**Figure 1.** TRPC3/6/7 channels make a detectable, but minor, contribution to the melanopsin photocurrent in M4 intrinsically photosensitive retinal ganglion cells (ipRGCs). (**A**) M4 ipRGC filled with Neurobiotin (Nb, magenta) in a Control retina and immunolabeled for the M4 marker SMI-32 (green). Right panel shows merged image of overlap, M4 cell outlined with white dotted circle. Scale bar 100 μm. (**B**) Example whole-cell voltage-clamp recording of M4 ipRGC photocurrent in a Control retina stimulated by a 50 ms, full-field 480 nm light ($6.08 \times 10^{15}$ photons · cm$^{-2}$ · s$^{-1}$) pulse in the presence of synaptic blockers. The trace is labeled with the Maximum, Early, Intermediate, and Late components used in the analysis (see 'Materials and methods'). (**C**) Individual light responses of Control (top row, black, n = 7) and TRPC3/6/7 KO (middle row, orange, n = 4) M4 ipRGCs. The bottom row are the overlaid averages of M4 Control (black) and TRPC3/6/7 KO (orange) light response. (**D**) The absolute values of the current amplitudes for light responses in panel (**C**) are quantified. The graphs for the Maximum, Early, Intermediate, and Late components compare the current amplitude for Control (black, n = 7) and TRPC3/6/7 KO (orange, n = 4) M4 ipRGCs. The Early component of TRPC3/6/7 KO M4 ipRGCs is significantly reduced compared to Control (*p=0.0424). Recordings are in the presence of synaptic blockers. All cells were stimulated with a 50 ms flash of blue (480 nm) light ($6.08 \times 10^{15}$ photons · cm$^{-2}$ · s$^{-1}$). *p<0.05. n.s., not significant. Analysis performed using the Mann–Whitney $U$ test (see 'Materials and methods'). Bars in (D) indicate mean.

The online version of this article includes the following source data and figure supplement(s) for figure 1:

**Source data 1.** Photocurrent components for Control and TRPC3/6/7 KO M4 intrinsically photosensitive retinal ganglion cells (ipRGCs).

**Figure supplement 1.** Control and TRPC3/6/7 KO M4s have similar capacitance and input resistance.

**Figure supplement 1—source data 1.** Capacitance and input resistance values for Control and TRPC3/6/7 KO M4 cells.

50 μM ZD7288 should eliminate the melanopsin-dependent M4 photocurrent. However, we observed no reduction in the M4 photocurrent in 50 μM ZD7288 despite elimination of the HCN tail current under these conditions (*Figure 2B and C*). Thus, though M4 ipRGCs do express HCN channels, HCN channels are not required for melanopsin phototransduction.

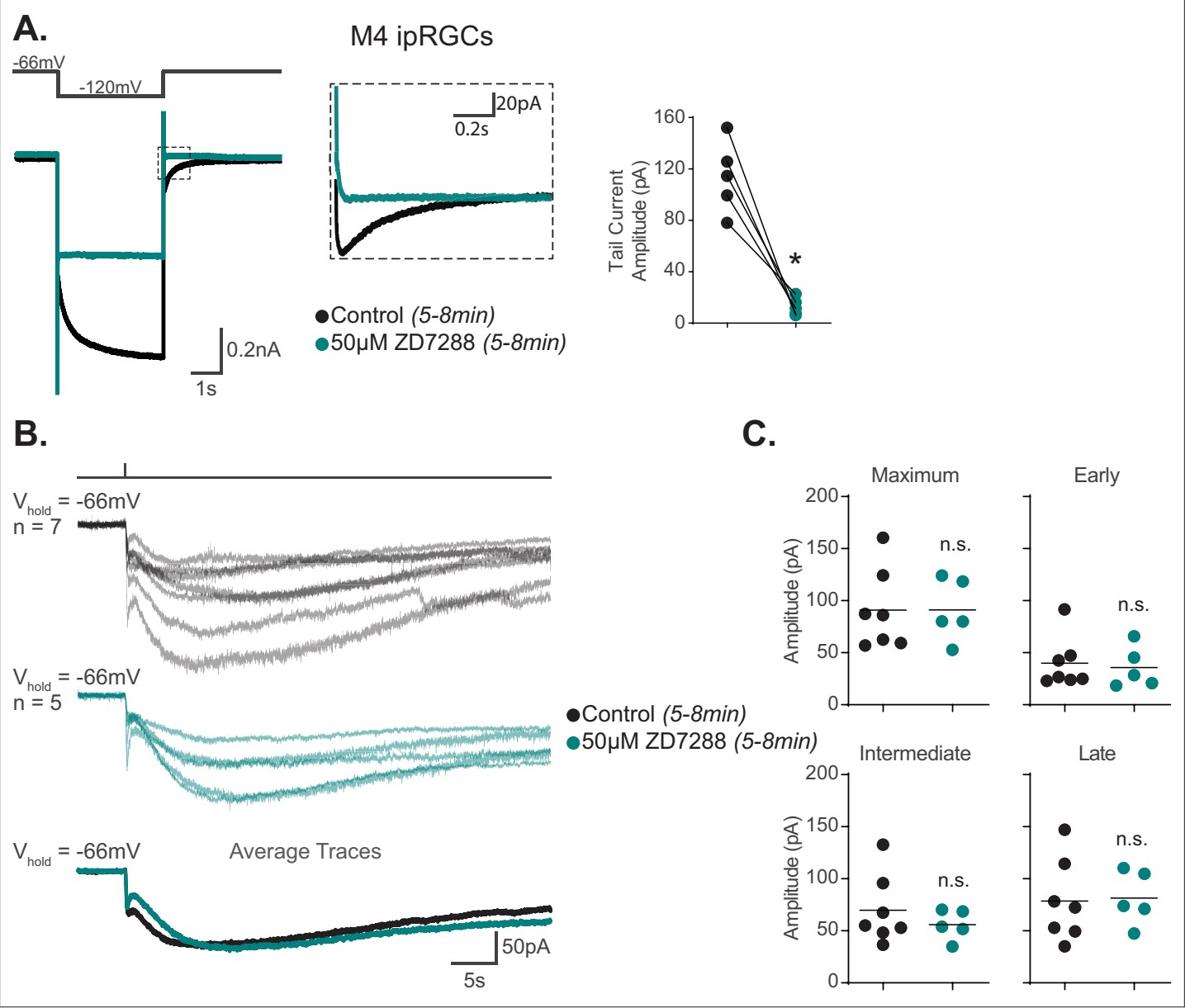

**Figure 2.** Hyperpolarization-activated cyclic nucleotide-gated (HCN) channels are not required for M4 phototransduction. (**A**) Left: representative recording a typical Control M4 intrinsically photosensitive retinal ganglion cell (ipRGC) (black), hyperpolarized from –66 mV to –120 mV and stepped back to the original holding potential. Cell was then incubated for 5–8 min with 50 μM ZD7288 (teal) and subjected to the same voltage-clamp protocol. Tail currents are boxed. Middle: magnified boxed tail currents. Right: absolute value of the tail current amplitude of Control M4 cells (black, n = 5) before and after application of 50 μM ZD7288 for 5–8 min (teal, n = 5). 50 μM ZD7288 for 5–8 min successfully blocked HCN-mediated tail currents of M4 ipRGCs (p=0.0079). Analysis performed using Wilcoxon signed-rank test (see 'Materials and methods'). (**B**) Individual light responses of M4 cells recorded in control solution (black, n = 7) or M4 cells incubated with 50 μM ZD7288 for 5–8 min (teal, n = 5, same cells for which tail current was quantified in panel **A**). Bottom row shows the overlaid average light response trace for Control (black) and 50 μM ZD7288 for 5–8 min (teal) M4 cells. (**C**) Maximum, Early, Intermediate, and Late absolute value amplitudes of Control (black, n = 7) M4 ipRGCs or cells exposed to 5–8 min of 50 μM ZD7288 (teal, n = 5). The photocurrent of M4 cells in 5–8 min of 50 μM ZD7288 is unaffected by blockade of HCN channels as shown in by the insignificant change in all the analyzed components. n.s., not significant. Analysis performed using the Mann–Whitney *U* test (see 'Materials and methods'). Bars in (C) indicate mean.

The online version of this article includes the following source data and figure supplement(s) for figure 2:

**Source data 1.** Hyperpolarization-activated cyclic nucleotide-gated (HCN) tail current and photocurrent components of M4 intrinsically photosensitive retinal ganglion cells (ipRGCs) in the 5–8 min of 50 μM ZD7288.

**Figure supplement 1.** The M4 hyperpolarization-activated cyclic nucleotide-gated (HCN) current and tail current are not modulated by light.

*Figure 2 continued on next page*

*Figure 2 continued*

**Figure supplement 1—source data 1.** M4 hyperpolarization-activated cyclic nucleotide-gated (HCN) current and tail current in dark vs. light.

**Figure supplement 2.** Prolonged application of ZD7288 reduces the M4 photocurrent via off-target effects.

**Figure supplement 2—source data 1.** Hyperpolarization-activated cyclic nucleotide-gated (HCN) tail current and photocurrent components of M4 intrinsically photosensitive retinal ganglion cells (ipRGCs) in the 20 min of 50 µM ZD7288.

As a second test of HCN involvement in M4 phototransduction, we investigated whether light exposure changes the amplitude of the HCN current or tail current in M4 ipRGCs. If melanopsin phototransduction opens HCN channels by shifting their voltage dependence, then in constant light at –66 mV (the starting potential in our HCN protocol) some proportion of HCN channels would already be open because we measure robust photocurrents at –66 mV (e.g. *Figures 1C and 2B*). It then follows that a hyperpolarizing step from –66 mV to –120 mV would evoke a smaller HCN current and tail current in light because there would be fewer HCN channels left to open compared to darkness when a larger pool of HCN channels are available to open with that same hyperpolarizing step. To test this, we compared the amplitude of the M4 HCN current and tail current in darkness and after 90 s of bright, background light (480 nm light at $6.08 \times 10^{15}$ photons $\cdot$ cm$^{-2}$ $\cdot$ s$^{-1}$). This background light evoked a steady inward current as evidenced by the increased holding current required at –66 mV in light versus dark (*Figure 2—figure supplement 1D* and see 'Materials and methods'). We found no change in either the M4 HCN current or tail current amplitudes (*Figure 2—figure supplement 1A–C*), indicating that light does not modulate the HCN current, further arguing against HCN involvement in M4 phototransduction.

The permeation properties of the M4 transduction channel(s) should be reflected in the photocurrent I–V relationship. Therefore, as a third test of HCN involvement in M4 phototransduction, we compared the reversal potential of the M4 HCN tail current versus that of the M4 photocurrent. We previously showed that the photocurrent I–V relationship of Control and TRPC3/6/7 KO M4 cells has a negative slope, reverses at the equilibrium potential for potassium channels (–90 mV), and drives an increase in input resistance and cellular excitability, indicating that potassium channels are the primary target of M4 phototransduction (*Sonoda et al., 2018*). However, the I–V relationship of the HCN tail current has not previously been reported for M4 ipRGCs. The reversal of the HCN tail current fit to a linear regression is used to estimate a reversal potential of HCN channels in other cell types and is usually between –25 mV and –50 mV (*Van Hook and Berson, 2010*). If melanopsin phototransduction does open HCN channels in M4 cells, then we would expect the extrapolated reversal potential of the HCN tail current to match the reported reversal potential of the M4 photocurrent. To test this, we measured the reversal potential of the M4 HCN tail current and compared it to that of the M4 photocurrent reported previously (*Sonoda et al., 2018*). To do this, we used a voltage-clamp protocol designed to open HCN channels and recorded the HCN tail current across multiple test potentials in the absence and then presence of 5–8 min of 50 µM ZD7288 (*Van Hook and Berson, 2010*; *Chen and Yang, 2007*; *Figure 3A*). We then calculated the HCN tail current amplitude by subtracting the tail current amplitude in ZD7288 from those generated in control solution at each test potential and plotted the ZD7288-sensitive current and used a linear fit to extrapolate the reversal potential (*Van Hook and Berson, 2010*; *Chen and Yang, 2007*; *Figure 3A and B*). Unlike the I–V relationship of the M4 photocurrent reversal at –90 mV, the linear fit of the M4 HCN tail current shows an extrapolated reversal at –26 mV. Though consistent with the properties of HCN channels (*Biel et al., 2009*), the extrapolated M4 HCN tail current I–V reversal is distinct from the reversal of the M4 photocurrent (M4 photocurrent replotted in Figure 3B from *Sonoda et al., 2018*, see also Figures 7D and S6 of *Sonoda et al., 2018*). These distinct differences further suggest that M4 phototransduction does not open HCN channels.

## Extended ZD7288 application drives off-target effects on the M4 photocurrent

Though *Jiang et al., 2018* reported complete blockade of the M4 photocurrent in 50 µM ZD7288, we saw no change in photocurrent amplitude in the same cells where we had confirmed complete blockade of HCN tail currents after 5–8 min incubation of ZD7288 (*Figure 2*; *Jiang et al., 2018*). We therefore sought to reconcile our experimental outcome with those previously described. ZD7288

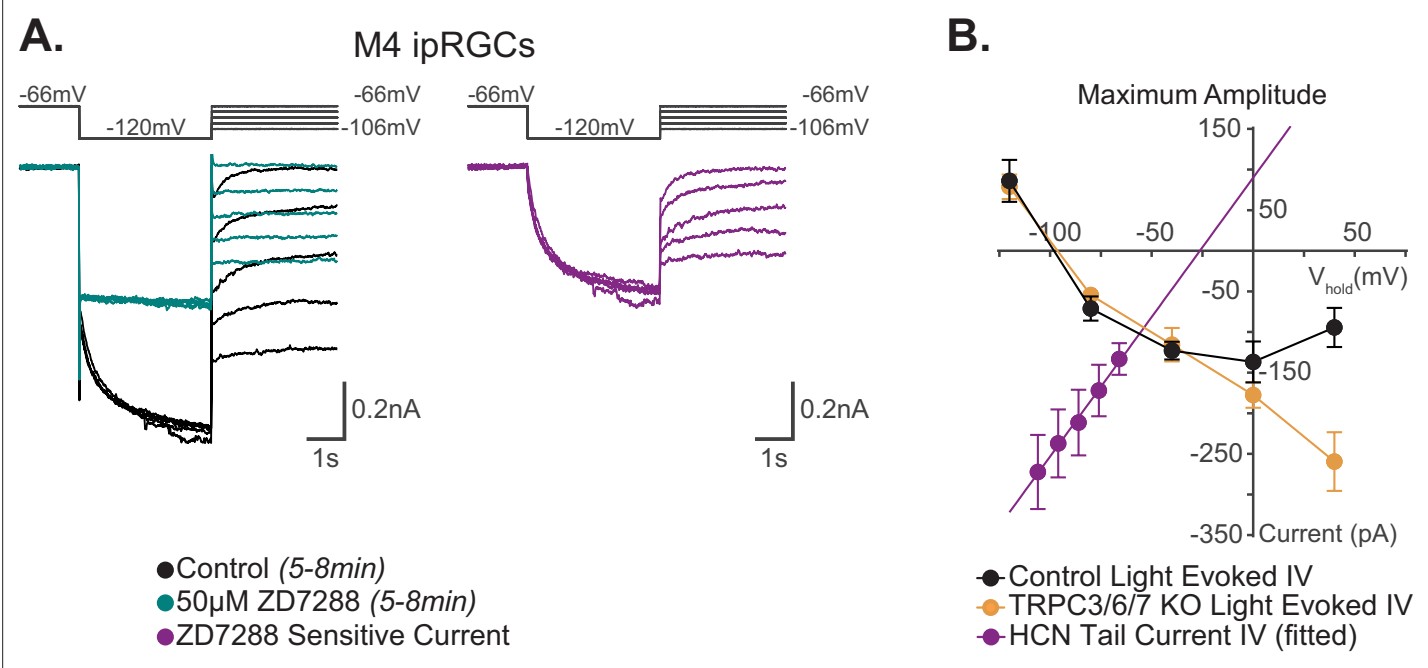

**Figure 3.** I–V relationship of M4 photocurrent reversal is distinct from extrapolated M4 hyperpolarization-activated cyclic nucleotide-gated (HCN) tail current reversal. (**A**) Left: example Control (black) M4 cell hyperpolarized to –120 mV to activate HCN channels followed by a step to various test potentials ranging from –106 mV to –66 mV. The voltage protocol is then repeated in the same cell following 50 μM ZD7288 for 5–8 min (teal). Right: the ZD-sensitive currents (magenta) are obtained by subtracting the application of 50 μM ZD7288 for 5–8 min (teal) from Control (black). (**B**) Current–voltage (I–V) relationship of M4 HCN tail current derived from the ZD-sensitive trace (magenta) in (**A**). A linear fit is used to extrapolate the reversal potential of the HCN tail current as described previously (*Chen and Yang, 2007*; *Van Hook and Berson, 2010*). The HCN-mediated tail current has a positive slope and reversed at –26 mV (magenta, n = 7). Data are represented as mean ± SEM. I–V relationship of HCN channels (magenta) in M4 intrinsically photosensitive retinal ganglion cells (ipRGCs) is compared to the 10 s light step ($10^{12}$ photons $\cdot$ cm$^{-2}$ $\cdot$ s$^{-1}$) I–V relationships of the maximum photocurrent of Control (black, n = 25, 5 cells/group) and TRPC3/6/7 KO M4 cells (orange, n = 23 cells, 4–6 cells/group) reported by *Sonoda et al., 2018*. Data are represented as mean ± SEM. The light-evoked I–V relationships for both Control (black) and TRPC3/6/7 KO (orange) M4 cells have a negative slope and reverse at –90 mV (see Figure 7D [10 s light step], and Figure 6 [100 ms light step] in *Sonoda et al., 2018*). This contrasts with the reversal of the calculated HCN tail current reversal of –26 mV (magenta) of M4 ipRGCs.

The online version of this article includes the following source data for figure 3:

**Source data 1.** I–V relationship of the M4 photocurrent and the I–V relationship of the M4 hyperpolarization-activated cyclic nucleotide-gated (HCN) tail current.

has been reported to have off-target effects on other ion channels (*Felix et al., 2003*; *Do and Bean, 2003*; *Sánchez-Alonso et al., 2008*; *Wu et al., 2012*). If the reported prior blockade of the M4 photocurrent by 50 μM ZD7288 was due to off-target effects, then longer incubation with 50 μM ZD7288 could result in off-target effects that mimic the previously observed reduction in the M4 photocurrent (ZD7288 incubation time was not reported in the prior study) (*Jiang et al., 2018*). To test this, we treated M4 ipRGCs with 50 μM ZD7288 for 20 min. Importantly, this longer application period caused no further reduction in the HCN tail currents compared to cells incubated for 5–8 min, indicating that the shorter 5–8 incubation time is sufficient for full HCN blockade (*Figure 2—figure supplement 2A and D*). Despite identical HCN channel blockade, this longer 20 min incubation with 50 μM ZD7288 now essentially abolished the M4 photocurrent and increased M4 cell input resistance compared to control (*Figure 2—figure supplement 2B–E*). These results are consistent with potential nonspecific blockade of potassium channels by ZD7288 and provide a methodological explanation for previous conclusions of HCN involvement in M4 phototransduction (*Do and Bean, 2003*; *Jiang et al., 2018*). Collectively, our results argue against a role for HCN channels in M4 phototransduction and provide support for our previous conclusions that melanopsin phototransduction results in closure of potassium channels with a minor contribution of TRPC3/6/7 channels.

## HCN channels are not required for M2 phototransduction

Previous work has reported that HCN and TRPC6/7 channels each contribute significantly to M2 phototransduction (*Jiang et al., 2018*). Considering the evidence against HCN involvement in M4 phototransduction, we next revisited the role of HCN channels in M2 phototransduction. We first sought to replicate previous findings that the HCN antagonist ZD7288 partially reduces the M2 photocurrent (*Jiang et al., 2018*). To do this, we performed whole-cell recordings of M2 ipRGCs identified under brief epifluorescent illumination in retinas of Opn4-GFP mice where only M1, M2, and M3 ipRGCs are labeled with EGFP (*Schmidt et al., 2008*). All cells were filled with Neurobiotin and identified as M2 ipRGCs post-recording using established criteria including ON stratification (M2 ipRGCs are the only ON type of ipRGC labeled in Opn4-GFP mice), large dendritic arbors, and lack of SMI-32 immunolabeling (*Figure 4A*; *Schmidt and Kofuji, 2009*; *Lee and Schmidt, 2018*; *Lucas and Schmidt, 2019*; see 'Materials and methods'). Notably, HCN tail currents of control M2 ipRGCs showed smaller, more variable HCN tail current amplitudes compared to those measured in M4 cells, consistent with previous reports (*Jiang et al., 2018*; *Figures 2A and 4B*). We then measured HCN tail currents in M2 cells before and after 5–8 min incubation with 50 µM ZD7288 and found that the HCN currents and tail currents were completely abolished under these conditions, identifying an effective concentration and incubation period of ZD7288 for full HCN channel blockade in M2 cells (*Figure 4B*). We next evaluated the role of HCN channels in M2 phototransduction. If HCN channels are involved in M2 phototransduction, then complete blockade of HCN channels via application of 50 µM ZD7288 for 5–8 min should significantly reduce the M2 melanopsin photocurrent (*Jiang et al., 2018*). To test this, we recorded the M2 photocurrent evoked by brief, full-field, 50 ms flashes of high photopic ($6.08 \times 10^{15}$ photons · cm$^{-2}$ · s$^{-1}$) 480 nm light in the presence or absence of 5–8 min of 50 µM ZD7288 (*Figure 4C–E*). M2 photocurrents in control cells consistently showed a transient peak in the Early phase of the response and then a smaller, slow component that persisted for more than 30 s following light offset (*Figure 4C and D*). As with M4 cells, HCN channel blockade with an effective concentration and incubation period of ZD7288 did not reduce the M2 photocurrent (*Figure 4D and E*), suggesting that HCN channels are not a target of melanopsin phototransduction in M2 cells. As a second test of HCN involvement in M2 phototransduction, we tested whether light altered the HCN current or tail current amplitude of M2 ipRGCs when stepped from –66 mV to –120 mV. Again, like M4 cells, we found no change in the amplitude of the M2 HCN tail current recorded in background light compared to darkness (*Figure 4—figure supplement 1A–C*), despite a steady, light-evoked inward current at the starting holding potential of –66 mV (*Figure 4—figure supplement 1D and E*). This indicates that the M2 HCN current is not modulated by light. This further supports the conclusion that HCN channels, though expressed in M2 cells, are not a target of M2 melanopsin phototransduction.

## Extended ZD7288 application drives off-target effects on the M2 photocurrent

Our findings suggest that HCN channels are not opened by melanopsin phototransduction in M2 ipRGCs. However, previous work did report partial blockade of the melanopsin photocurrent in M2 ipRGCs following incubation with ZD7288 (*Jiang et al., 2018*). Given our findings that ZD7288 incubation can have off-target effects that decrease the M4 photocurrent (*Figure 2—figure supplement 2*), we postulated that prolonged exposure to high concentrations of ZD7288 may likewise reduce the M2 photocurrent via off-target effects (*Felix et al., 2003*; *Do and Bean, 2003*; *Sánchez-Alonso et al., 2008*; *Wu et al., 2012*). To test this, we measured the photocurrent of M2 ipRGCs following 20 min incubation with 50 µM ZD7288. Despite no additional reduction in HCN current, this longer incubation period resulted in an ~50% reduction in the maximum amplitude of the M2 photocurrent, consistent with the previously reported reduction (*Figure 4—figure supplement 2A–D*; *Jiang et al., 2018*). These findings suggest that this additional photocurrent blockade observed in this study and in previous work was due to off-target effects on non-HCN channels in M2 cells.

## TRPC channels are a major phototransduction target in M2 ipRGCs

Given the lack of HCN involvement in M2 phototransduction, we next sought to identify the channels involved. TRPC6/7 have been reported to contribute to the M2 photocurrent (*Jiang et al., 2018*; *Perez-Leighton et al., 2011*). We therefore sought to determine the contribution of TRPC3/6/7 channels to M2 phototransduction. To do this, we recorded the M2 photocurrent in TRPC3/6/7KO; Opn4-GFP

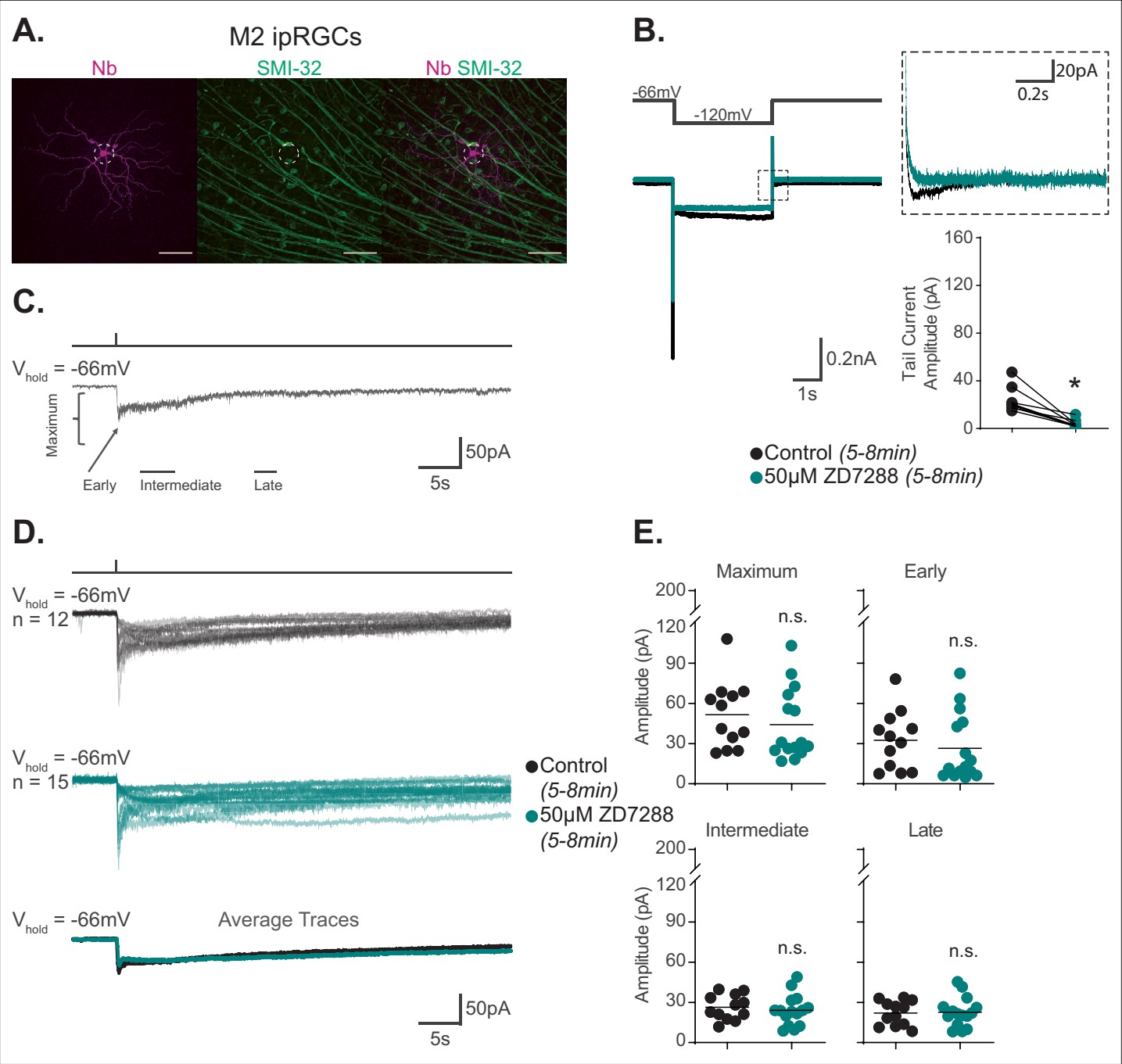

**Figure 4.** Hyperpolarization-activated cyclic nucleotide-gated (HCN) channels are not required for M2 phototransduction. (**A**) M2 intrinsically photosensitive retinal ganglion cell (ipRGC) filled with Neurobiotin (Nb, magenta) in a Control retina and immunolabeled for the M4 marker SMI-32 (green). Right panel shows merged image and lack of immunolabeling of filled cell with SMI-32, confirming identity as an M2 ipRGC. Scale bar 100 µm. (**B**) Left: representative recording a typical Control M2 ipRGC (black), hyperpolarized from –66 mV to –120 mV and stepped back to the original holding potential. Cell was then incubated for 5–8 min with 50 µM ZD7288 (teal) and subjected to the same voltage-clamp protocol. Tail currents are boxed. Middle: magnified boxed tail currents. Right: absolute value of HCN tail current amplitude of Control M2 cells (black, n = 9) before and after application of 50 µM ZD7288 for 5–8 min (teal, n = 9). 50 µM ZD7288 for 5–8 min successfully blocked HCN tail currents of M4 ipRGCs (p=0.0039). Performed statistical analysis with Wilcoxon signed-rank test (see 'Materials and methods'). (**C**) Example light response of Control M2 ipRGC to a 50 ms, 480 nm light pulse (6.08 × 10$^{15}$ photons · cm$^{-2}$ · s$^{-1}$) in the presence of synaptic blockers. Trace is labeled with the Maximum, Early, Intermediate, and Late components used in the analysis (see 'Materials and methods'). (**D**) Individual light responses of Control (black, n = 12) M2 cells and M2 cells incubated with 50 µM ZD7288 for 5–8 min (teal, n = 15). Bottom row shows the overlaid average light response trace for Control (black) and 50 µM ZD7288 for 5–8 min (teal) M2 cells. (**E**) Absolute value of photocurrent amplitudes quantified for cells in (**C**). Photocurrent of M2 cells in 50 µM ZD7288 for 5–8 min is

*Figure 4 continued on next page*

*Figure 4 continued*

unaffected despite full blockade of HCN channels shown in (**B**). Performed statistical analysis with Mann–Whitney *U* test (see 'Materials and methods'). n.s., not significant. Bars in (E) indicate mean.

The online version of this article includes the following source data and figure supplement(s) for figure 4:

**Source data 1.** Hyperpolarization-activated cyclic nucleotide-gated (HCN) tail current and photocurrent components of Control M2 intrinsically photosensitive retinal ganglion cells (ipRGCs) in the 5–8 min of 50 µM ZD7288.

**Figure supplement 1.** M2 hyperpolarization-activated cyclic nucleotide-gated (HCN) current and tail current are not modulated by light.

**Figure supplement 1—source data 1.** Control M2 hyperpolarization-activated cyclic nucleotide-gated (HCN) current and tail current in dark vs. light.

**Figure supplement 2.** Prolonged application of ZD7288 reduces the M2 photocurrent via off-target effects.

**Figure supplement 2—source data 1.** Hyperpolarization-activated cyclic nucleotide-gated (HCN) tail current and photocurrent components of M2 intrinsically photosensitive retinal ganglion cells (ipRGCs) in the 20 min of 50 µM ZD7288.

---

retinas. We found that the maximum amplitude was reduced by ~75% in TRPC3/6/7 KO M2 ipRGCs, and the Early component was essentially completely abolished (*Figure 5A and B*). This was not due to changes in intrinsic properties because capacitance and resistance were identical in TRPC3/6/7 KO and Control M2 Cells (*Figure 5—figure supplement 1*). Additionally, we observed no further reduction in the small remaining photocurrent of TRPC3/6/7 KO M2 following 5–8 min incubation with the HCN antagonist ZD7288 (50 µM) (*Figure 5—figure supplement 2A–C*). Furthermore, both the HCN tail current and inward current were not significantly altered by light exposure in TRPC3/6/7 KO M2 ipRGCs (*Figure 5—figure supplement 3A and B*). Thus, as with WT M2 phototransduction, we find no evidence of HCN involvement in TRPC3/6/7 KO M2 ipRGCs.

Our data show that TRPC channels are a major target of melanopsin phototransduction in M2 ipRGCs (*Figure 5A and B*), but the full I–V relationship of the M2 photocurrent has never been reported. We therefore sought to generate I–V curves for the Maximum, Early, Intermediate, and Late components of M2 phototransduction. To do this, we measured the photocurrent of M2 ipRGCs at multiple holding potentials from –106 mV to +34 mV following stimulation with brief, full-field, 50 ms flashes of high photopic ($6.08 \times 10^{15}$ photons · cm$^{-2}$ · s$^{-1}$) 480 nm light (*Figure 6A and B*, *Figure 6—figure supplement 1*). In control M2 ipRGCs, the inward current was the largest at –106 mV (*Figure 6A*, *Figure 6—figure supplement 1A*) and decreased at more depolarized holding potentials (*Figure 6A*, *Figure 6—figure supplement 1A*). The light-evoked I–V curves for each component had a positive slope, consistent with an increase in conductance via channels opening (*Figure 6B*). All photocurrent components in control M2 ipRGCs reversed between +10 mV and +25 mV, suggesting contributions from cation channels that are more permeable to sodium or calcium, such as TRPC channels (*Figure 6B*). In TRPC3/6/7 KO retinas, I–V relationships reversed between 0 mV and +10 mV, and the Maximum, Intermediate, and Late component amplitudes were reduced at more hyperpolarized, physiologically relevant, potentials while the Early component was reduced at all potentials (*Figure 6*, *Figure 6—figure supplement 1B*). Notably, HCN tail currents in M2 ipRGCs reversed at –32 mV, which is distinct from that of the photocurrent reversal at +19 mV for control and ~0 mV for TRPC3/6/7 KO M2 cells (*Figure 7A and B*), further arguing against HCN involvement in M2 phototransduction.

## Voltage-gated calcium channels are a major target of melanopsin phototransduction in M2 ipRGCs

M1 and M2 ipRGCs heavily rely on TRPC channels for phototransduction, a commonality that would predict similar photocurrent I–V relationships for these subtypes. However, we observed that the I–V relationship of M1 and M2 ipRGCs differed in shape and reversal potential, with M1 cells having a more linear I–V relationship that reverses at 0 mV compared to a more complex I–V relationship that reverses +19 mV for M2 cells (*Figure 8A*; *Warren et al., 2006*; *Hartwick et al., 2007*; *Graham et al., 2008*; *Xue et al., 2011*; *Perez-Leighton et al., 2011*; *Sonoda et al., 2018*). This observation suggests that while both M1 and M2 cells require TRPC channels for melanopsin phototransduction, another type of channel may be interacting with TRPC channels in M2 ipRGCs to shift the reversal of the I–V relationship. Calcium entry through voltage-gated calcium channels (VGCCs) would shift the reversal potential of the photocurrent I–V to more positive voltages. Interestingly, VGCCs, like TRPC channels, can be modulated by Gq (*Bloomquist et al., 1988*; *Scott et al., 1995*; *Niemeyer et al., 1996*; *Bertaso et al., 2003*; *Panda et al., 2005*; *Qiu et al., 2005*; *Hildebrand et al., 2007*; *Warren et al.,*

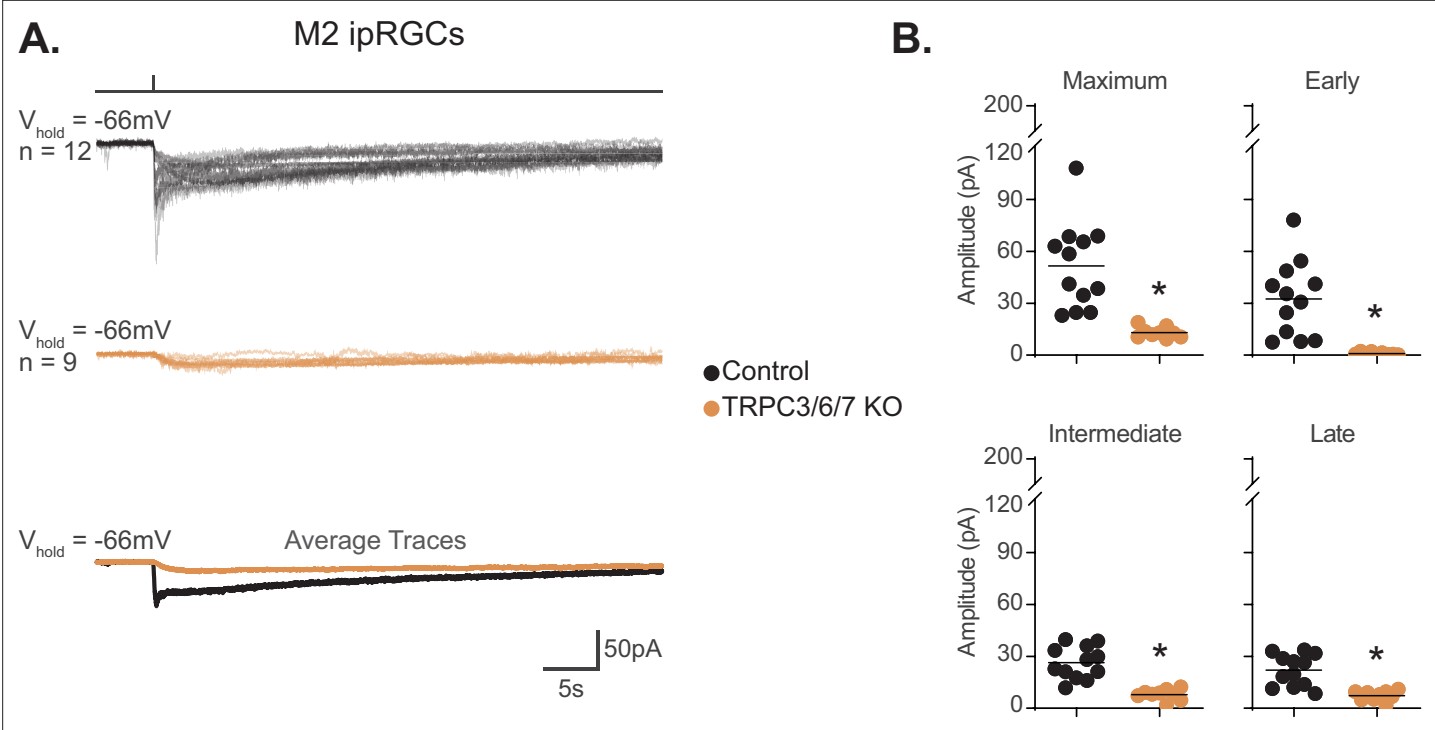

**Figure 5.** TRPC3/6/7 channels are a major phototransduction target in M2 intrinsically photosensitive retinal ganglion cells (ipRGCs). (**A**) Individual photocurrent recordings of Control (Opn4-GFP, black, n = 12) and TRPC3/6/7 KO (Opn4-GFP; TRPC 3/6/7 KO, orange, n = 9) M2 ipRGCs to a 50 ms, 480 nm light pulse ($6.08 \times 10^{15}$ photons $\cdot$ cm$^{-2} \cdot$ s$^{-1}$) in the presence of synaptic blockers. Bottom row shows the overlaid average light response trace for Control (black) and TRPC3/6/7 KO (orange). (**B**) Absolute value of photocurrent amplitudes quantified for cells in (**A**). The photocurrent was significantly reduced for all components in TRPC3/6/7 KO M2 cells (*p<0.0001). Analysis performed with Mann–Whitney *U* test (see 'Materials and methods'). Bars in (B) indicate mean.

The online version of this article includes the following source data and figure supplement(s) for figure 5:

**Source data 1.** Photocurrent components for Control and TRPC3/6/7 KO M2 intrinsically photosensitive retinal ganglion cells (ipRGCs).

**Figure supplement 1.** Control and TRPC3/6/7 KO M2s have similar capacitance and input resistance.

**Figure supplement 1—source data 1.** Capacitance and input resistance values for Control and TRPC3/6/7 KO M2 cells.

**Figure supplement 2.** Blockade of hyperpolarization-activated cyclic nucleotide-gated (HCN) channels does not reduce the TRPC3/6/7 KO M2 photocurrent.

**Figure supplement 2—source data 1.** Hyperpolarization-activated cyclic nucleotide-gated (HCN) tail current and photocurrent components of TRPC3/6/7 KO M2 intrinsically photosensitive retinal ganglion cells (ipRGCs) in the 5–8 min of 50 μM ZD7288.

**Figure supplement 3.** Hyperpolarization-activated cyclic nucleotide-gated (HCN) current and tail current are not modulated by light in M2 intrinsically photosensitive retinal ganglion cells (ipRGCs) lacking TRPC3/6/7 channels.

**Figure supplement 3—source data 1.** TRPC3/6/7KO M2 hyperpolarization-activated cyclic nucleotide-gated (HCN) current and tail current in dark vs. light.

*2006*; *Graham et al., 2008*; *Xue et al., 2011*; *Keum et al., 2014*; *Sonoda et al., 2018*; *Jiang et al., 2018*). Moreover, in other cell types several TRPC subunits have been shown to interact with VGCCs (*Soboloff et al., 2005*; *Onohara et al., 2006*; *Yan et al., 2009*; *Perissinotti et al., 2021*). Therefore, we hypothesized that VGCCs may be a target of M2 phototransduction. If VGCCs are required for M2 phototransduction, then we would expect VGCC antagonists to reduce the M2 photocurrent. To test this, we recorded the photocurrent of M2 ipRGCs in control solution and in the presence of a cocktail of VGCC antagonists (see 'Materials and methods'; *Figure 8B and C*). Application of VGCC antagonists resulted in a significant reduction of the overall M2 photocurrent similar to that seen in TRPC3/6/7 KO M2 cells, indicating that VGCCs are required for M2 melanopsin phototransduction (*Figure 8B and C*). Importantly, VGCC blockade did not reduce the M1 photocurrent (*Figure 8D and E*). This indicates that VGCCs are not required for melanopsin phototransduction in M1 ipRGCs and

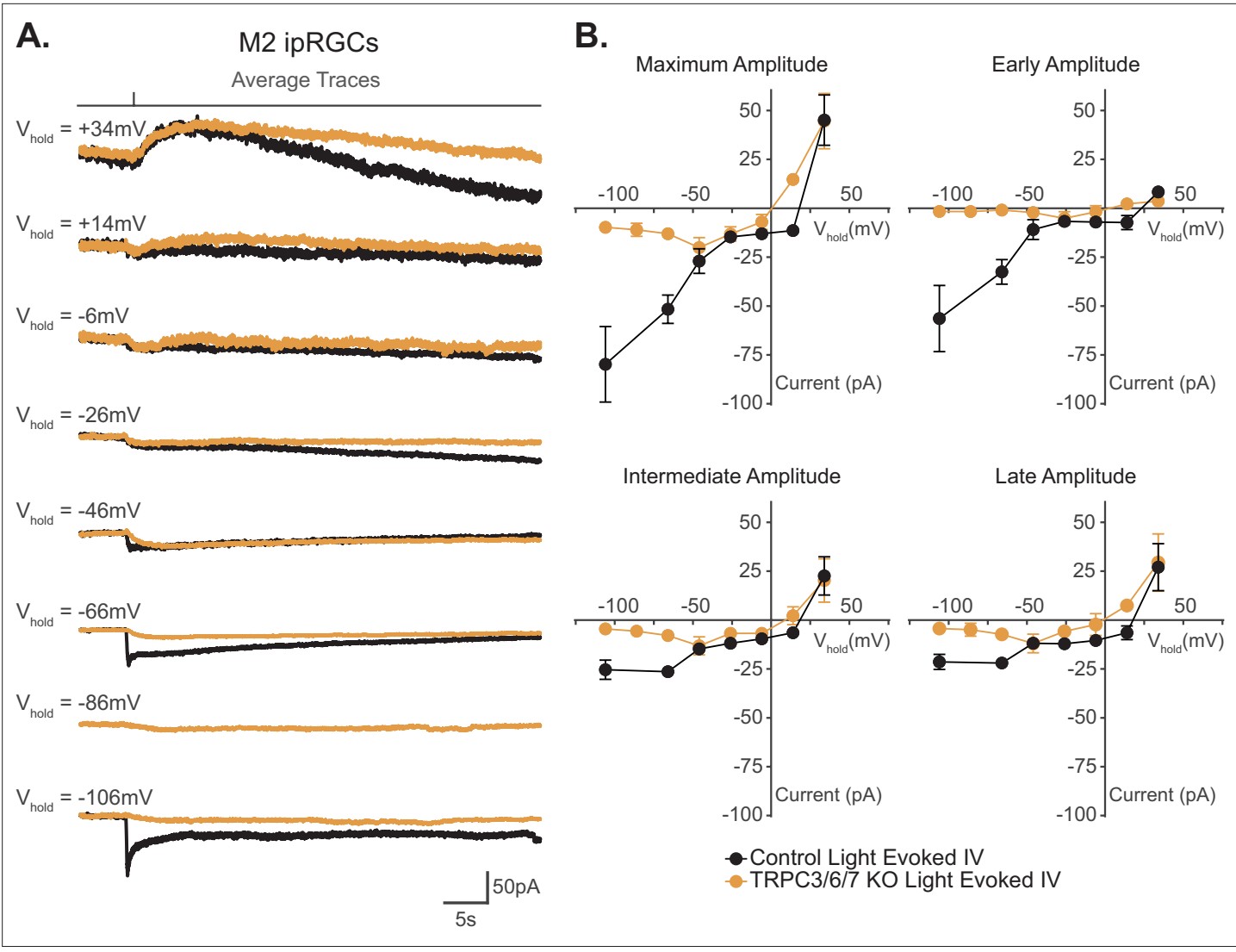

**Figure 6.** I–V relationship of M2 photocurrent. (**A**) Average photocurrent traces for Control (black) and TRPC3/6/7 KO (orange) M2 intrinsically photosensitive retinal ganglion cells (ipRGCs) at various holding potentials (–106 mV to +34 mV) to a 50 ms, 480 nm light pulse (6.08 × 10$^{15}$ photons · cm$^{-2}$ · s$^{-1}$) in the presence of synaptic blockers. Individual traces for all cells are shown in *Figure 6—figure supplement 1*. (**B**) Photocurrent amplitudes for all Control (black, n = 52, 4–12 cells/group) and TRPC3/6/7 KO (orange, n = 43, 2-9 cells/group) M2 ipRGCs at various holding potentials for Maximum, Early, Intermediate, and Late components. I–V relationships for Control M2 cells reverse between +10–25 mV and have a positive slope. Data are represented as mean ± SEM.

The online version of this article includes the following source data and figure supplement(s) for figure 6:

**Source data 1.** I–V relationship of M2 photocurrent for Control and TRPC3/6/7 KO cells.

**Figure supplement 1.** Individual M2 photocurrents used to plot the I–V relationship.

that VGCCs may account for some of the observed differences in the M1 versus M2 photocurrent I–V relationships.

## T-type VGCCs interact with TRPC channels in M2 ipRGCs

T-type VGCCS are one potential candidate target of the M2 phototransduction cascade. T-type VGCC mRNA expression has been reported in M2 ipRGCs (*Figure 9—figure supplement 1A*; *Tran et al., 2019*), and we detect T-type VGCC immunolabeling in ON-stratifying ipRGCs in retinal sections (*Figure 9—figure supplement 1B*). To test this functionally, we performed voltage clamp recordings of M2 cells stepping membrane voltage from –70 mV to more depolarized voltages (see 'Materials and methods' for detailed protocol) and compared current amplitudes in control solution and after

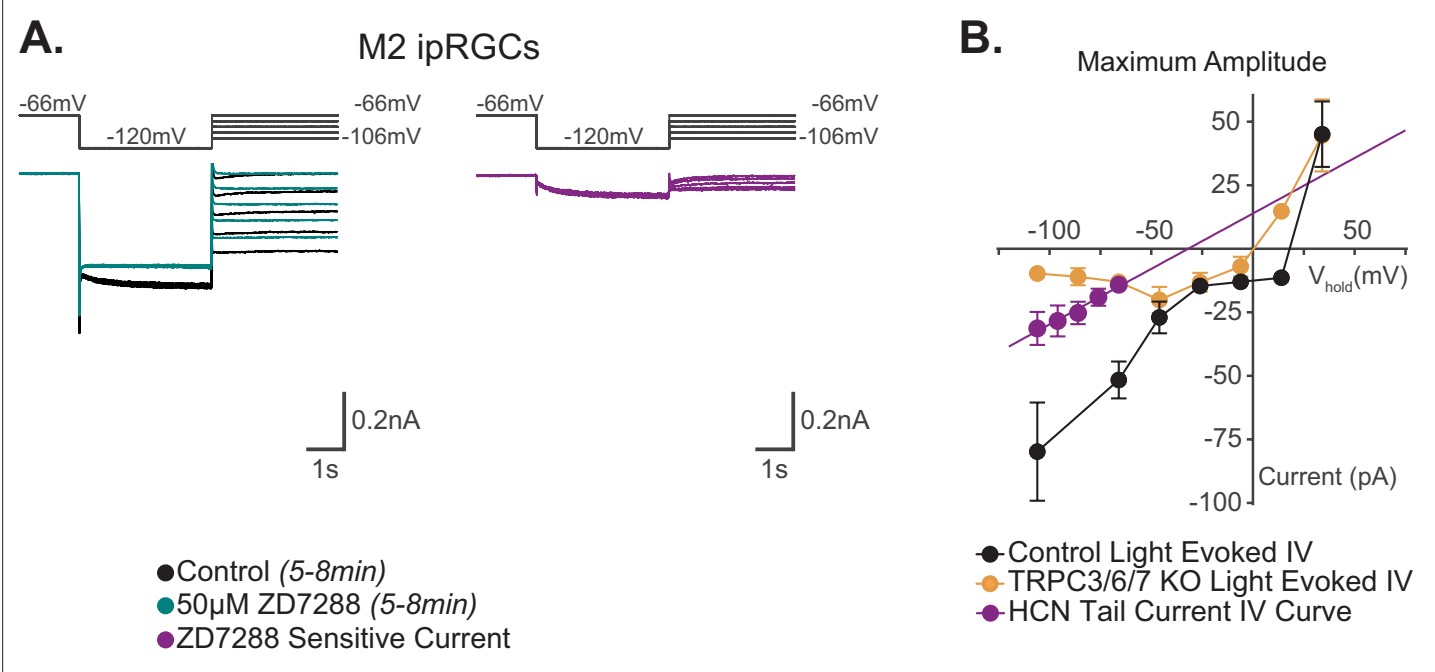

**Figure 7.** Reversal potential of M2 photocurrent is distinct from extrapolated reversal of M2 hyperpolarization-activated cyclic nucleotide-gated (HCN) tail current. (**A**) Left: example Control (black) M2 cell hyperpolarized to –120 mV to activate HCN channels followed by a step to various test potentials ranging from –106 mV to –66 mV. The voltage protocol was then repeated in the same cell following 50 µM ZD7288 for 5–8 min (teal). Right: the ZD-sensitive currents (magenta) are obtained by subtracting the application of 50 µM ZD7288 for 5–8 min (teal) from Control (black). (**B**) I–V relationship of M2 HCN tail current derived from the ZD-sensitive trace (magenta) in (**A**) and M2 photocurrent I–V curves derived from the maximum component amplitudes for Control (black, n = 52, 4–12 cells/group) and TRPC3/6/7 KO (orange, n = 43, 2–9 cells/group) cells from *Figure 6B*. A linear fit was used to extrapolate a reversal potential of –32 mV for HCN tail currents (magenta, n = 5), which is distinct from the photocurrent reversal of Control M2 ipRGCs of +19 mV. All data are represented as mean ± SEM.

The online version of this article includes the following source data for figure 7:

**Source data 1.** I–V relationship of the M2 photocurrent compared to the I–V relationship of the M2 hyperpolarization-activated cyclic nucleotide-gated (HCN) tail current.

application of the selective T-type VGCC antagonist TTA-P2 (10 µM) (*Figure 9A*, *Figure 9—figure supplement 2*; *Wu et al., 2018*; *Randall and Tsien, 1997*; *Timic Stamenic et al., 2019*; *Zhang et al., 2023*; *Dreyfus et al., 2010*; *Choe et al., 2011*). We found that TTA-P2 reduced M2 current amplitudes in this protocol, and that TTA-P2-sensitive currents showed I–V relationships consistent with T-type VGCC channels (*Wu et al., 2018*; *Perez-Reyes, 2003*; *Randall and Tsien, 1997*; *Monteil et al., 2000*; *Fox et al., 1987*; *Dreyfus et al., 2010*; *Choe et al., 2011*). Combined, these data provide evidence that M2 ipRGCs express functional T-type VGCC channels.

We next tested the hypothesis that T-type VGCCs are gated by the melanopsin phototransduction cascade in M2 cells. If T-type VGCCs are required for M2 melanopsin phototransduction, then blockade of T-type VGCCs should reduce or eliminate the M2 photocurrent. To test this, we recorded the M2 photocurrent under blockade of T-type VGCCs with the specific T-type antagonist TTA-P2 (10 µM). We found that T-type blockade nearly eliminates the M2 photocurrent without reducing the HCN tail current (*Figure 9B and C*, *Figure 9—figure supplement 3*). The M2 photocurrent was similarly reduced in the presence of a second T-type VGCC antagonist, mibefradil (10 µM) (*Figure 10*), suggesting that T-type VGCCs are required for melanopsin phototransduction. Importantly, mibefradil had no effect on the M2 photocurrent in TRPC3/6/7 KO M2 cells (*Figure 10*), suggesting that TRPC 3/6/7 channels and T-type channels act through a common pathway. We next recorded the M2 photocurrent in a cocktail of VGCC antagonists that did not contain T-type antagonists (see 'Materials and methods'). This VGCC cocktail lacking T-type antagonists did not significantly reduce the M2 photocurrent (*Figure 10—figure supplement 1*), further supporting a role for T-type VGCCs in M2 phototransduction. As with the full VGCC cocktail used above, TTA-P2 alone had no effect on the M1 photocurrent (*Figure 9D and E*), suggesting that VGCCs are dispensable for M1 phototransduction.

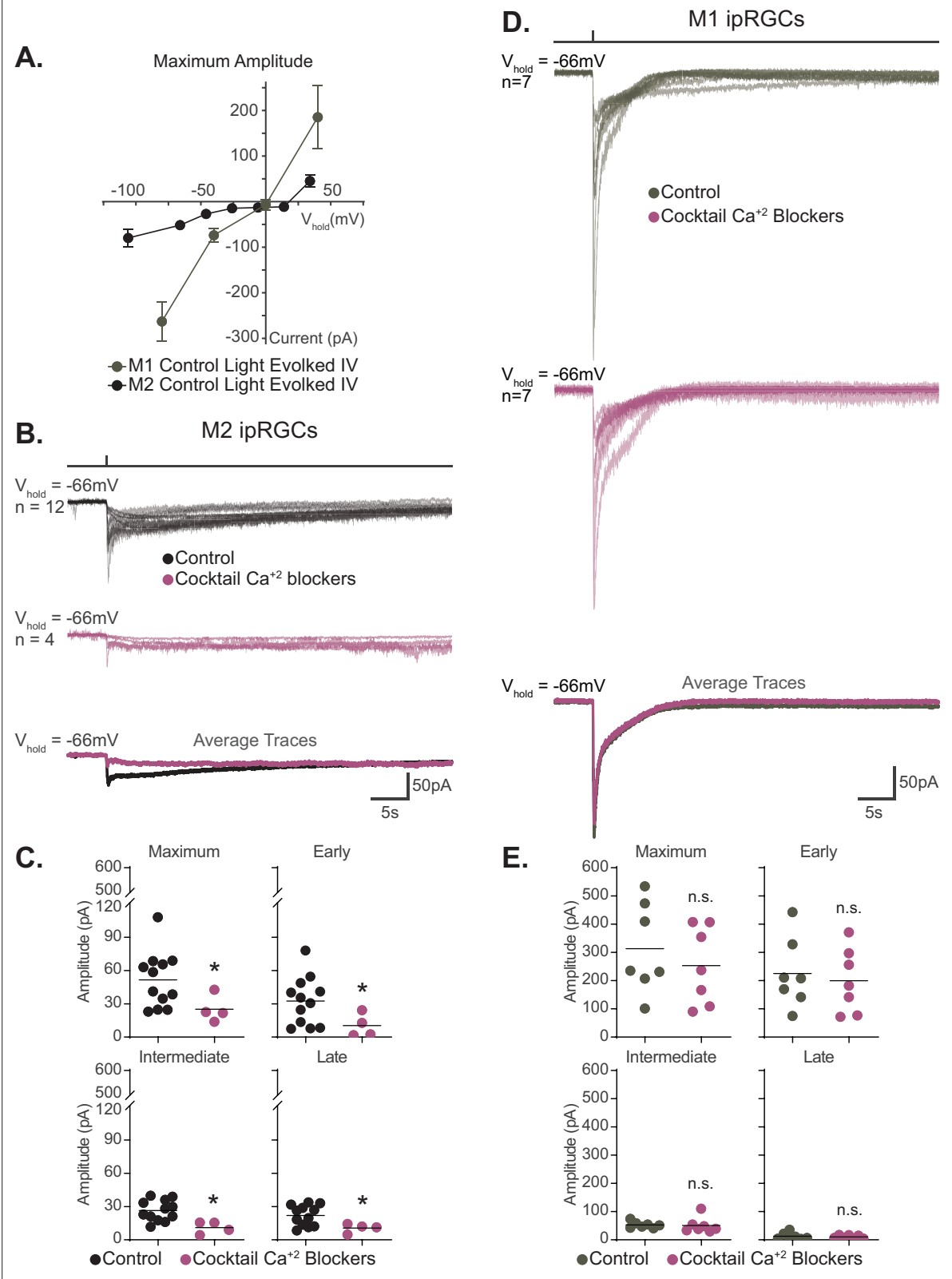

**Figure 8.** Voltage-gated calcium channels are required for M2, but not M1, phototransduction. (**A**) Light-evoked I–V relationships for the maximum amplitude of the photocurrent in Control M2 (black, n = 52, 4–12 cells/group, from *Figure 6B*) and M1 (gray, n = 19, 4–6 cells/group, replotted from *Sonoda et al., 2018*) intrinsically photosensitive retinal ganglion cells (ipRGCs). Control M2 ipRGCs (black) reversal is +19 mV compared to 0 mV in Control M1 cells (gray), despite major contribution of TRPC channels to M1 and M2 phototransduction. M1 photocurrents are in response to a

*Figure 8 continued on next page*

*Figure 8 continued*

10 second blue (480 nm) light step ($10^{12}$ photons · $cm^{-2}$ · $s^{-1}$) reported in **Sonoda et al., 2018**. Data are represented as mean ± SEM. (**B**) Photocurrent of Control (black, n = 12) M2 ipRGCs in synaptic blocker cocktail and M2 ipRGCs incubated in a cocktail also containing voltage-gated calcium channel (VGCC) antagonists (plum, n = 4). Bottom row: overlaid average light response traces for Control cells (black) and cells in a cocktail of calcium blockers (plum). VGCC antagonists were dissolved in the synaptic blocker cocktail and consisted of: 10 µM nifedipine, 5 µM nimodipine, 400 nM $\omega$-agatoxin IVA, 3 µM $\omega$-conotoxin GVIA, 3 nM SNX-482, and 10 µM mibefradil dihydrochloride. (**C**) Absolute value of the photocurrent of cells in (**B**) are quantified. The Maximum (p=0.0297), Early (p=0.0418), Intermediate (p=0.0044), and Late (p=0.0418) component amplitudes of the photocurrent are significantly reduced in the M2 ipRGCs in the presence of a cocktail of calcium blockers compared to Control. (**D**) Photocurrent of Control (gray, n = 7) M1 ipRGCs in synaptic blocker cocktail and M1 ipRGCs incubated in a cocktail also containing VGCC antagonists (plum, n = 7). Bottom row: overlaid average light response traces for Control cells (black) and cells in a cocktail of calcium blockers (plum). VGCC antagonists were dissolved in the synaptic blocker cocktail and consisted of 10 µM nifedipine, 5 µM nimodipine, 400 nM $\omega$-agatoxin IVA, 3 µM $\omega$-conotoxin GVIA, 3 nM SNX-482, and 10 µM mibefradil dihydrochloride. (**E**) Absolute value of photocurrent for cells in (**D**) is quantified. There is no significant difference in any of the photocurrent components in M1 cells treated with a cocktail of calcium blockers. All recordings for M1 and M2 ipRGCs were made in Control (Opn4-GFP) retinas in response to a 50 ms flash of blue (480 nm) light (6.08 × $10^{15}$ photons · $cm^{-2}$ · $s^{-1}$) and in the presence of synaptic blockers. *p<0.05. n.s., not significant. Performed statistical analysis with Mann–Whitney *U* test (see 'Materials and methods'). Bars in (C,E) represent mean.

The online version of this article includes the following source data for figure 8:

**Source data 1.** M1 and M2 photocurrent components in a cocktail of voltage-gated calcium channel antagonists.

Importantly, this complete lack of blockade of the M1 photocurrent by TTA-P2 or the VGCC antagonist cocktail also argues against reduction of the M2 photocurrent resulting from nonspecific effects of these drugs on TRPC channels. Collectively, these findings suggest a key, previously unappreciated, role for T-type VGCCs in M2, but not M1, melanopsin phototransduction.

## Discussion

Collectively, our findings identify previously unknown components of the melanopsin phototransduction cascade in M2 ipRGCs and reconcile multiple models of melanopsin phototransduction across ipRGC subtypes (**Figure 11**). While melanopsin phototransduction in M1 ipRGCs opens TRPC 3/6/7 channels, in M2 ipRGCs melanopsin phototransduction requires both TRPC and T-type VGCCs, and in M4 ipRGCs melanopsin phototransduction closes leak potassium channels. These subtypes vary in their roles in behavior, their intrinsic properties, and their integration of light information from intrinsic and retinal pathways. These varying mechanisms likely optimize each subtype for their role in behavior. For example, in M4 ipRGCs, closure of potassium channels increases the excitability of M4 cells, enhancing cellular (and likely behavioral) contrast sensitivity while M1 ipRGCs perform simpler luminance-detecting functions to impact behaviors that integrate luminance information across long timescales (**Berson et al., 2002**; **Hattar et al., 2002**; **Ruby et al., 2002**; **Lucas et al., 2003**; **Wong et al., 2005**; **Güler et al., 2008**; **Perez-Leighton et al., 2011**; **Wong, 2012**; **Schmidt et al., 2014**; **Emanuel and Do, 2015**; **Sonoda et al., 2018**). Interestingly, the role of M2 ipRGCs in behavior has not been resolved. A key future question will be understanding how the M2 phototransduction cascade impacts cellular signaling to influence M2-driven behaviors.

Using rigorous, established ipRGC subtype identification criteria, we find no role for HCN channels in M2 or M4 ipRGCs. Though both subtypes express HCN channels, the melanopsin photocurrent in each subtype is insensitive to HCN blockade, has a distinct I–V relationship to that of HCN channels, and is not modulated by light. In our previous work, we plotted the I–V relationship of M4 ipRGCs across three experimental paradigms using (1) whole-cell patch-clamp recordings to 10 s light stimuli, (2) whole-cell patch-clamp recordings to 100 ms light stimuli, or (3) nucleated patch recordings to 10 s light stimuli (**Sonoda et al., 2018**). In all cases, the I–V relationship of the melanopsin photocurrent in WT and TRPC3/6/7 KO M4 cells had a negative slope that reversed at the potassium equilibrium potential, consistent with decreasing conductance (channel closure) through potassium channel closure via melanopsin phototransduction (**Sonoda et al., 2018**). Moreover, the reversal of this I–V relationship could be shifted to a newly calculated potassium equilibrium potential when the external potassium concentration was altered, further supporting a role for potassium channels in M4 phototransduction (**Sonoda et al., 2018**). Our previous work found evidence for a minor contribution of TRPC channels in photopic light (**Sonoda et al., 2018**), which we replicate in our findings here at even higher illumination. In this study, we show that the reversal potential of the M4 melanopsin photocurrent at –90 mV is distinct from that of the M4 HCN tail current at –26 mV.

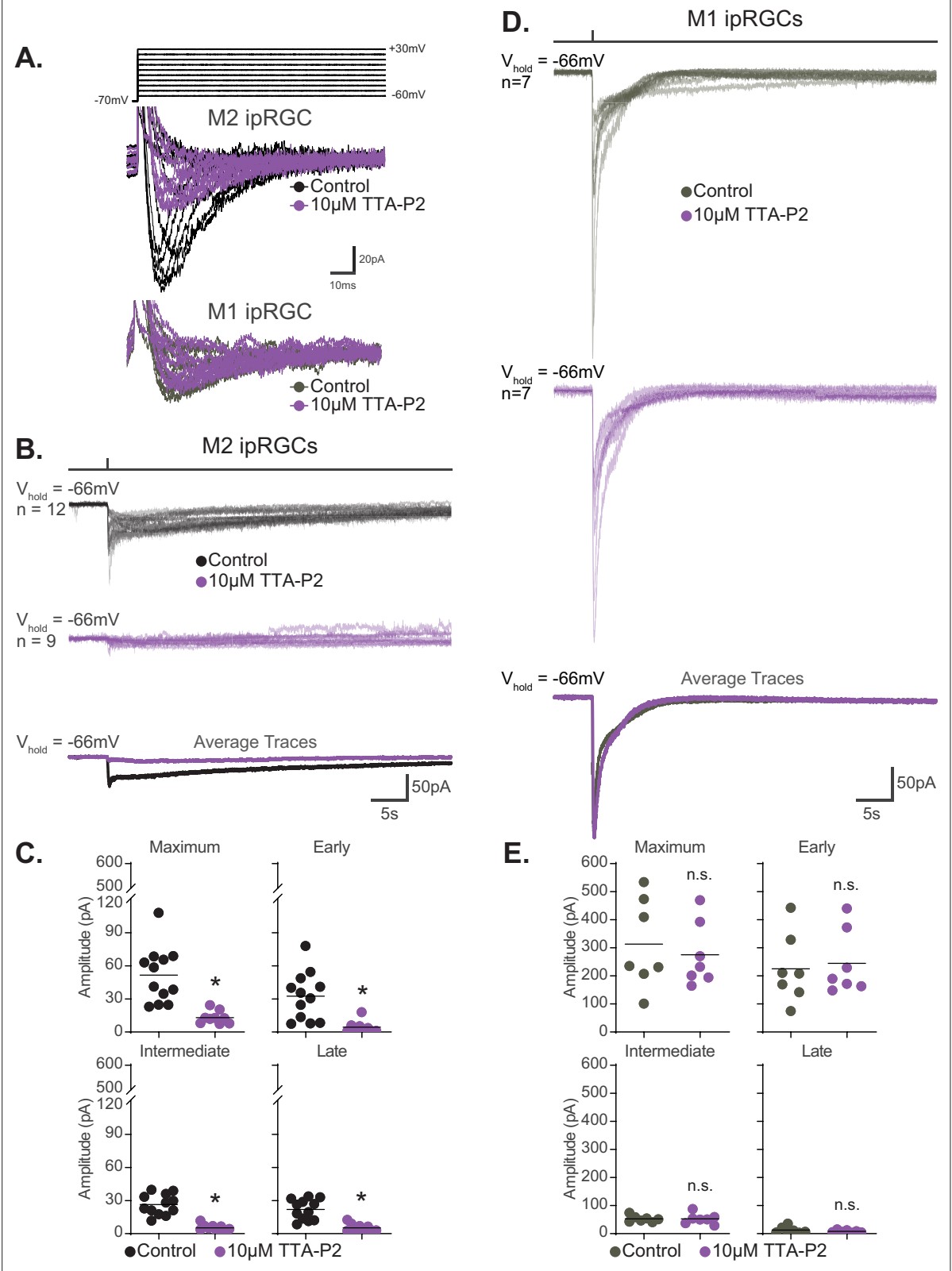

**Figure 9.** T-type voltage-gated calcium channels are required for M2, but not M1, phototransduction. (**A**) Top: calcium currents were isolated from Control M2 intrinsically photosensitive retinal ganglion cells (ipRGCs) (black, n = 4) held at –70 mV and depolarized to multiple voltage steps from –60 mV to +30 mV. Cells were subsequently exposed to 5 min of the T-type voltage-gated calcium currents 10 µM TTA-P2 (purple, n = 4) to block T-type voltage-gated channels followed by the same voltage step commands. Bottom: calcium currents were recorded from Control M1 ipRGCs (gray, n =

*Figure 9 continued on next page*

*Figure 9 continued*

2) and then in the presence of 10 µM TTA-P2 (purple, n = 2). (**B**) Individual light responses of Control (black, n = 12) M2 cells and M2 cells incubated with 10 µM TTA-P2 (purple, n = 9). Bottom row shows the overlaid average light response trace for Control (black) and 10 µM TTA-P2 (purple) M2 cells. (**C**) Absolute value of photocurrent amplitudes quantified for cells in (**B**). The photocurrent of M2 cells in 10 µM TTA-P2 is significantly reduced for all components (*p<0.0001) when T-type voltage calcium channels are blocked shown in (**A**). (**D**) Individual light responses of Control (gray, n = 7) M1 cells and M1 cells incubated with 10 µM TTA-P2 (purple, n = 7). Bottom row shows the overlaid average light response trace for Control (gray) and 10 µM TTA-P2 (purple) M1 cells. (**E**) Absolute value of photocurrent amplitudes quantified for cells in (**D**). Photocurrent of M1 cells in 10 µM TTA-P2 is unaffected. All recordings for M1 and M2 ipRGCs were made in Control (Opn4-GFP) retinas in response to a 50 ms flash of blue (480 nm) light (6.08 × $10^{15}$ photons · $cm^{-2}$ · $s^{-1}$) and in the presence of synaptic blockers. * p<0.05. n.s., not significant. Performed statistical analysis with Mann–Whitney *U* test (see 'Materials and methods'). Bars in (C,E) represent mean.

The online version of this article includes the following source data and figure supplement(s) for figure 9:

**Source data 1.** Photocurrent components for M1 and M2 intrinsically photosensitive retinal ganglion cells (ipRGCs) in the presence of the T-type voltage-gated calcium channel antagonist, TTA-P2.

**Figure supplement 1.** T-type voltage-gated calcium channels (VGCCs) are expressed in intrinsically photosensitive retinal ganglion cells (ipRGCs).

**Figure supplement 2.** M2 intrinsically photosensitive retinal ganglion cells (ipRGCs) have T-type currents.

**Figure supplement 2—source data 1.** M1 and M2 calcium currents in the presence of TTA-P2.

**Figure supplement 3.** TTA-P2 does not block hyperpolarization-activated cyclic nucleotide-gated (HCN) channels.

**Figure supplement 3—source data 1.** M2 hyperpolarization-activated cyclic nucleotide-gated (HCN) tail currents in the presence of TTA-P2.

Additionally, we report that the HCN antagonist ZD7288 fails to block the M4 photocurrent at a concentration and incubation period (5–8 min at 50 µM) that fully eliminates the HCN tail current. Additionally, light does not alter the amplitude of the M4 HCN current. Importantly, we were able to replicate previous findings that ZD7288 blocks the M4 photocurrent, but only after a long, 20 min incubation time. This longer incubation time did not further reduce HCN current but did increase the M4 cell input resistance, suggesting that the M4 photocurrent blockade following longer incubations was due to off-target effects of ZD7288, which have been reported previously in other systems (*Do and Bean, 2003*; *Felix et al., 2003*; *Sánchez-Alonso et al., 2008*; *Wu et al., 2012*). Thus, while M4 ipRGCs clearly *express* HCN channels, we find no evidence that HCN channels are in fact *gated* by melanopsin phototransduction. Our findings in TRPC3/6/7 KO M4 cells also confirm that TRPC3/6/7 channels play a minor role in M4 phototransduction in bright light, which we had shown previously in lower photopic light levels (*Sonoda et al., 2018*), arguing against a previous report that there is no effect on M4 photocurrent amplitude in TRPC knockout M4 cells (*Jiang et al., 2018*). Because much of the early transient component of the photocurrent, which depends heavily on TRPC channels, was not detected in previous work (potentially due to bleaching following epifluorescent localization), it is possible that this contribution of TRPC channels to M4 phototransduction was simply missed under highly light adapted conditions recording conditions (*Jiang et al., 2018*).

HCN channels had also been reported to play a role in M2 phototransduction (*Jiang et al., 2018*), and in light of our findings in M4 cells, we next examined the role of HCN channels in M2 phototransduction. Similar to our findings in M4 ipRGCs, 5–8 min incubation period with ZD7288 effectively blocked M2 HCN channels but failed to reduce the M2 photocurrent in either Control or TRPC3/6/7 KO cells. Likewise, the HCN tail current showed a calculated reversal potential at a voltage distinct from the M2 photocurrent and was unaffected by light. Each of these findings argues against HCN involvement in M2 phototransduction. Only with the longer, 20 min incubation period with ZD7288 were we able to replicate the previously reported ~50% reduction in M2 ipRGCs (*Jiang et al., 2018*). As with M4 cells, this blockade occurred despite no additional reduction in the HCN current, supporting the conclusion that this additional photocurrent blockade was due to off-target effects on non-HCN channels (*Do and Bean, 2003*; *Felix et al., 2003*; *Sánchez-Alonso et al., 2008*; *Wu et al., 2012*). Of note, ZD7288 has also been shown to block T-type calcium channels via off-target blockade, providing further support for this interpretation (*Sánchez-Alonso et al., 2008*; *Wu et al., 2012*). It is important to note that we could not achieve light levels as high as those used in *Jiang et al., 2018*, leaving open the possibility that HCN channels play a role at very high light levels. However, this possibility seems unlikely because the high photopic light levels used in this study are saturating and generate maximum photocurrents, yet we are unable to detect any contribution from HCN channels

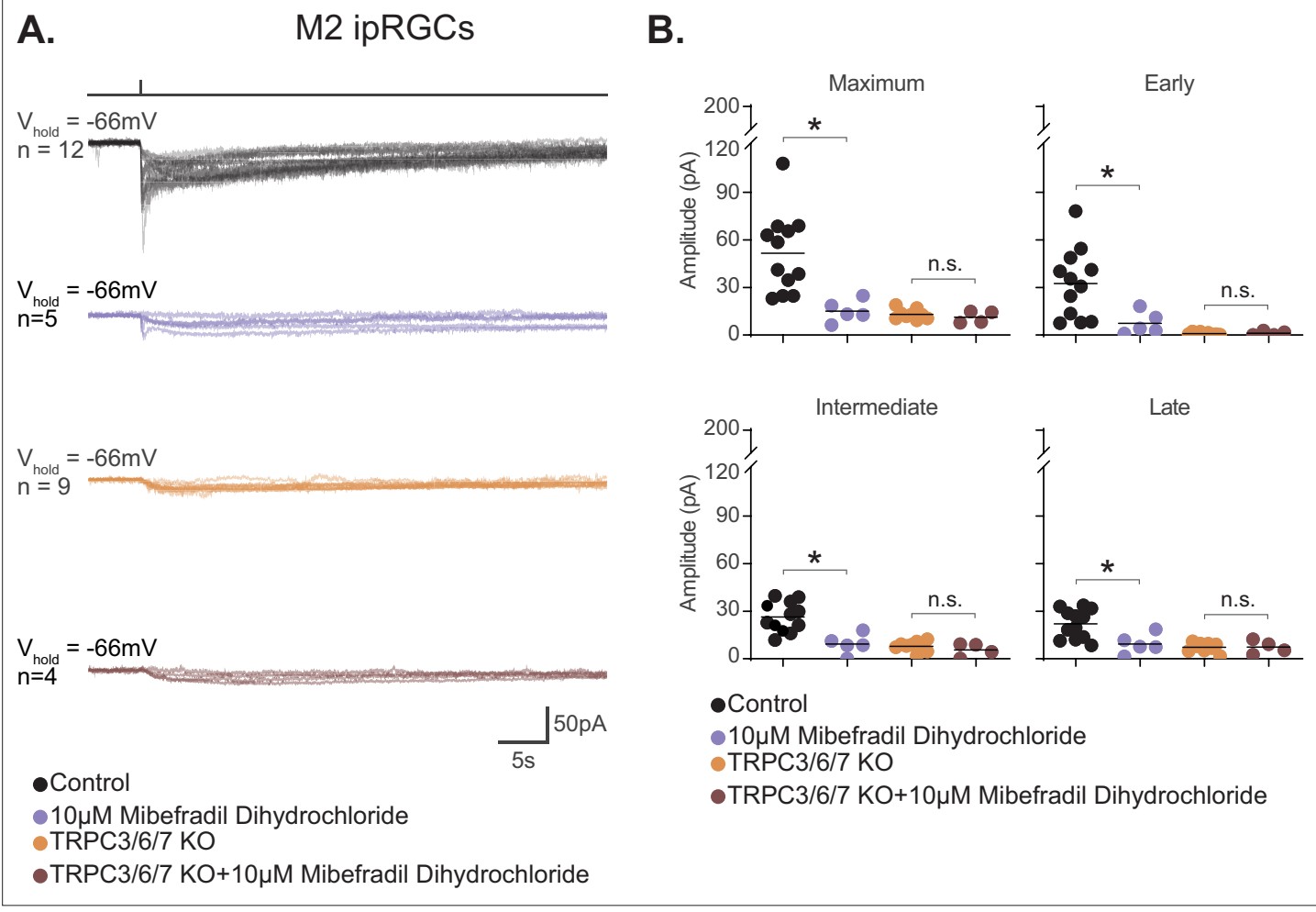

**Figure 10.** M2 photocurrent is blocked by a second T-type voltage-gated calcium channel (VGCC) antagonist, mibefradil dihydrochloride. (**A**) Photocurrent of Control (black, n = 12) and TRPC3/6/7 KO (orange, n = 9) M2 cells recorded in synaptic blockers alone. Control and TRPC3/6/7KO M2 cells incubated in the T-type VGCC antagonist 10 μM mibefradil dihydrochloride. Control (lilac, n = 5) and TRPC3/6/7 KO (brown, n = 4) M2 cells in synaptic blockers plus 10 μM mibefradil dihydrochloride. Cells were stimulated with a 50 ms flash of blue (480 nm) light ($6.08 \times 10^{15}$ photons $\cdot$ cm$^{-2}$ $\cdot$ s$^{-1}$). (**B**) Absolute value of photocurrent amplitudes for cells recorded in (**A**). The Maximum (p=0.0023), Early (p=0.0136), Intermediate (p=0.0023), and Late (p=0.0136) component amplitudes of the photocurrent are significantly reduced in the M2 intrinsically photosensitive retinal ganglion cells (ipRGCs) in the presence of 10 μM mibefradil dihydrochloride (lilac, n = 5). No further reduction of photocurrent was observed when TRPC3/6/7 M2 cells were incubated with 10 μM mibefradil dihydrochloride, indicating that TRPC3/6/7 channels and T-type VGCCs are acting in the same pathway. Bars in (**B**) represent mean.

The online version of this article includes the following source data and figure supplement(s) for figure 10:

**Source data 1.** Photocurrent components for Control and TRPC3/6/7 KO M2 cells exposed to a second T-type voltage-gated calcium channel antagonist, mibefradil dihydrochloride.

**Figure supplement 1.** Blockade of non-T-type voltage-gated calcium channels (VGCCs) does not reduce M2 photocurrent.

**Figure supplement 1—source data 1.** Photocurrent components for M2 intrinsically photosensitive retinal ganglion cells (ipRGCs) in a cocktail of voltage-gated calcium channel (VGCC) antagonist except for T-type VGCC antagonist.

for either M2 or M4 phototransduction. Collectively, our work suggests that HCN channels are not involved in propagating melanopsin light information to downstream processing centers in the brain.

Our reevaluation of melanopsin signaling in M2 ipRGCs prompted us to look at the role of TRPC channels in M2 phototransduction. Previous studies have noted that knockout of TRPC 6 or multiple TRPC3/6/7 subunits, as well as pharmacological inhibition of TRPC channels, causes 50% or greater reduction in the M2 photocurrent, suggesting that TRPC channels are a major transduction channel in M2 ipRGCs (*Perez-Leighton et al., 2011*; *Jiang et al., 2018*). In line with previous work, we found

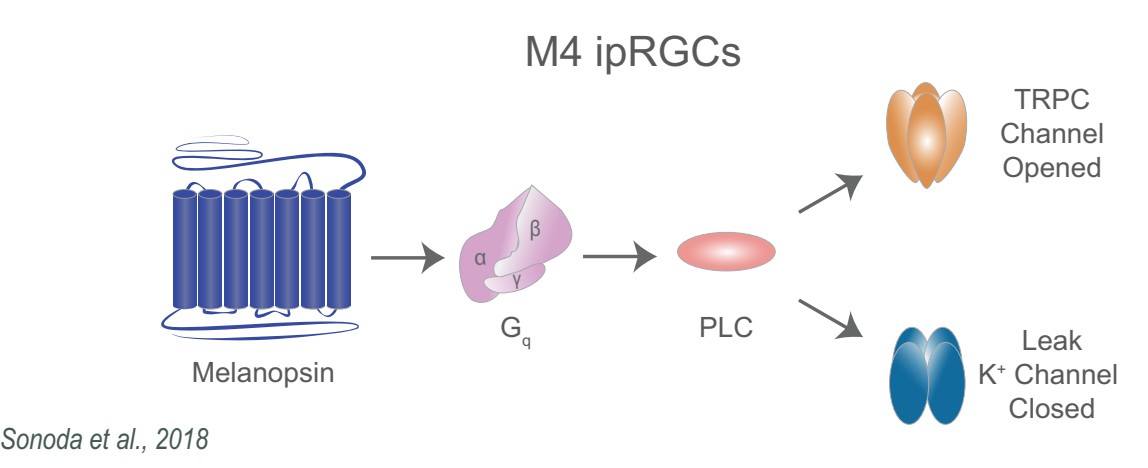

**Figure 11.** Diverse melanopsin phototransduction pathways in intrinsically photosensitive retinal ganglion cell (ipRGC) subtypes. Diagram depicting melanopsin phototransduction in M1, M2, and M4 ipRGCs updated from previous work based on current findings. Figure adapted from Figure 3 of *Contreras et al., 2021* (*Warren et al., 2006*; *Graham et al., 2008*; *Xue et al., 2011*; *Jiang et al., 2018*; *Sonoda et al., 2018*; *Perez-Leighton et al., 2011*).

that genetic elimination of TRPC subunits 3, 6, and 7, caused an ~75% reduction in their maximum photocurrent, suggesting that TRPC channels are a major melanopsin phototransduction channel in M2 cells.

Intriguingly, we observed that though both M1 and M2 ipRGCs require TRPC channels for the majority (M2) or entirety (M1) of their photocurrent, the I–V curves of M1 and M2 cells differed substantially, prompting us to probe whether additional unidentified transduction channels may be involved in one or both of these phototransduction pathways. We found that blockade of T-type VGCCs with either mibefradil or TTA-P2 resulted in an ~75% reduction in M2 photocurrent but had no effect on the M1 photocurrent, suggesting that T-type VGCCs are necessary for M2, but not M1, phototransduction. Importantly, this lack of blockade in M1 ipRGCs argues against any nonspecific effects of any VGCC blockers on TRPC channels. One possible mechanism for T-type VGCC activation is through modulation of channel excitability by Gq, the heterotrimeric G-protein activated in the melanopsin phototransduction cascade (*Sonoda et al., 2018*; *Chen et al., 2023*). Indeed, previous studies have shown that both TRPC channels and VGCCs can be modulated by Gq signaling (*Bloomquist et al., 1988*; *Scott et al., 1995*; *Niemeyer et al., 1996*; *Bertaso et al., 2003*; *Panda et al., 2005*; *Qiu et al., 2005*; *Warren et al., 2006*; *Hildebrand et al., 2007*; *Graham et al., 2008*; *Xue et al., 2011*; *Keum et al., 2014*; *Sonoda et al., 2018*; *Jiang et al., 2018*; *Zhang et al., 2021*). Additionally, it has recently been shown that cAMP, which likewise modulates T-type VGCC function through shifting its voltage-dependency (*Perez-Reyes, 2003*), may act downstream of Gq in M4 melanopsin phototransduction (*Chen et al., 2023*). If cAMP is activated by Gq in M2 cells, this is another potential mechanism for T-type VGCC activation (*Kim et al., 2006*; *Louiset et al., 2017*).

Of note, the similar degree of M2 photocurrent reduction by the T-type antagonists mibefradil and TTA-P2 to that seen in TRPC3/6/7 KO M2 cells, combined with lack of TRPC3/6/7 KO M2 cell photocurrent reduction in mibefradil, suggests that TRPC channels and T-type VGCCs act through a common phototransduction pathway. It is, however, noteworthy that a small transient component remained in control M2 cells under T-type blockade that was not present in TRPC3/6/7 KO M2 cells (*Figure 10A*), suggesting that some current through TRPC channels may remain under T-type channel blockade in Control M2 cells. This would suggest that T-type VGCCs may be downstream of TRPC channels. In other systems and cell types, TRPC channels have been shown to form complexes with and/or modulate the function of both L-type and T-type VGCCs (*Soboloff et al., 2005*; *Onohara et al., 2006*; *Yan et al., 2009*; *Perissinotti et al., 2021*). Moreover, T-type VGCCs open in response to small membrane depolarizations and have been shown to generate a sustained inward current at hyperpolarized voltages of in the range of –70 to –40 mV in other cell types due to incomplete inactivation (*Bijlenga et al., 2000*; *Perez-Reyes, 2003*). Thus, it is possible that the small TRPC3/6/7-mediated photocurrent in M2 ipRGCs depolarizes the membrane sufficiently to open T-type VGCCs and could generate a sustained M2 photocurrent at physiological voltages. This would align with the small transient current remaining under T-type blockade that is absent in TRPC3/6/7 KO M2 cells, but further testing is necessary to confirm this. Additionally, following strong depolarizations, T-type VGCCs are ideally situated to promote the high-frequency firing observed in M2 ipRGCs (*Schmidt and Kofuji, 2009*; *Goetz et al., 2022*) due to the fast activation and inactivation rate (*Perez-Reyes, 2003*). The mechanisms through which T-type VGCCs and TRPC3/6/7 channels interact with each other and other components of the melanopsin cascade remain a key question for future studies.

Our work adds to a growing literature describing the interaction between T-type calcium channels and TRPC channels, and supports divergent, TRPC-dependent phototransduction mechanisms in M1 and M2 ipRGCs. Additionally, this work resolves a key discrepancy in our understanding of M4 phototransduction and supports an emerging framework for melanopsin signaling that suggests that melanopsin phototransduction is tuned to optimize how ipRGC subtypes signal to influence their associated light-driven behaviors (*Figure 11*).

## Materials and methods

### Contact for reagent and resource sharing

Requests for reagents and resources should be directed to the lead contact, Tiffany Schmidt (tiffany.schmidt@northwestern.edu).

## Animals

All procedures were approved by the Animal Care and Use Committee at Northwestern University.

Both male and female mice were used with a mixed B6/129 background. All mice were between 30 and 90 d of age. For M4 cell recordings, we used WT and *Trpc3-/-* (*Hartmann et al., 2008*; RRID:MGI:3810154); *Trpc6-/-* (*Dietrich et al., 2005*; RRID:MGI:3623137); *Trpc7-/-* (*Perez-Leighton et al., 2011*; RRID:MGI:5296035) mice. For M2 and M1 cell recordings, we used *Opn4*-GFP (*Schmidt et al., 2008*) and *Opn4*-GFP; *Trpc3-/-*; *Trpc6-/-*; *Trpc7-/-* mice. For slice immunohistochemistry, we used *Opn4*cre/+ (*Ecker et al., 2010*; RRID:MGI:5285910) mice.

## Intravitreal injections

Mice were anesthetized with isoflurane delivered by Neurostar stereotaxic equipment (Robot Stereotaxic) and fixed to a nose cone under a surgical microscope. A 30-gauge needle was used to puncture a hole in the sclera, and each eye was injected with 1 μl of AAV2-hSyn-DIO-hM3D(Gq)-mCherry (6 × $10^{12}$ vg/ml; Addgene, Cat#44361-AAV2; RRID:Addgene_44361) using a custom Hamilton syringe with a 33-gauge needle (Borghuis Instruments). Mice were euthanized and the eyes were removed 4 weeks after injection.

## Ex vivo retina preparation for electrophysiology

All mice were dark-adapted overnight and euthanized by $CO_2$ asphyxiation followed by cervical dislocation.

Eyes were enucleated, and retinas were dissected under dim red light in carbogenated (95% $O_2$-5% $CO_2$) Ames' medium (Sigma-Aldrich). Retinas were then sliced in half and incubated in carbogenated Ames' medium at 26°C for at least 30 min. Retinas were then mounted on a glass-bottom recording chamber and anchored using a platinum ring with nylon mesh (Warner Instruments). The retina was maintained at 30–32°C and perfused with carbogenated Ames' medium at a 2–4 ml/min flow.

## Solutions for electrophysiology

All recordings were made in Ames' medium with 23 mM sodium bicarbonate. Synaptic transmission was blocked with 100 μM DNQX (Tocris), 20 μM L-AP4 (Tocris), 100 μM picrotoxin (Sigma-Aldrich), and 20 μM strychnine (Sigma-Aldrich) in Ames' medium. Then, 500 nM tetrodotoxin (TTX) citrate (Tocris) was added to the synaptic blocker solution for voltage-clamp experiments. For whole-cell recordings, the internal solution (*Jiang et al., 2018*) used contained 120 mM K-gluconate, 5 mM NaCl, 4 mM KCl, 10 mM HEPES, 2 mM EGTA, 4 mM ATP-Mg, 0.3 mM GTP-Na$_2$ and 7-Phosphocreatine-Tris, with the pH adjusted to 7.3 with KOH. The internal solution was passed through a sterile filter with a 0.22 μm pore size (Sigma-Aldrich). Prior to recording, 0.3% Neurobiotin (Vector Laboratories) and 10 μM Alexa Fluor 594 (Thermo Fisher) were added to internal solution. The HCN antagonist, ZD7288 (Tocris), was dissolved in distilled water and added to the synaptic blockers for a final concentration of 50 μM. 50 μM ZD7288 was bath applied for 5–8 min (an incubation period we identified to fully block the HCN tail current, *Figure 2A*) with minimum off-target effects, or for 20 min, which led to additional non-specific effects on the photocurrent of M2 and M4 ipRGCs (*Figure 2—figure supplement 2* and *Figure 4—figure supplement 2*). The cocktail of calcium (VGCC) blockers was added to the synaptic blockers and consisted of L-type blockers: 10 μM nifedipine and 5 μM nimodipine; P/Q-type blocker: 400 nM ω-agatoxin IVA; N-type blocker: 3 μM ω-conotoxin GVIA; R-type blocker: 3 nM SNX-482; and T-type blocker: 10 μM mibefradil dihydrochloride. Nifedipine (Tocris) and nimodipine (Tocris) were dissolved in DMSO and diluted in synaptic blockers. Mibefradil dihydrochloride (Tocris), SNX-482 (Tocris), ω-conotoxin GVIA (Tocris), and ω-agatoxin IVA (Tocris) were reconstituted in water and dilute in synaptic blockers to reach the final concentration. VGCC blockers were applied for 5 min to minimize off-target effects. To record calcium currents (*Figure 9*), a cesium based internal was used to block potassium and HCN channels. The cesium-based internal solution used contained 125 mM Cs-methanesulfonate, 10 mM CsCl, 1 mM MgCl$_2$, 5 mM EGTA, 10 mM Na-HEPES, 2 mM Na$_2$-ATP, 0.5 mM Na-GTP, and 10 mM Phosphocreatine. Cells were recorded in the presence of synaptic blockers with 2.5 mM TEA to block additional potassium channels.

## Light stimulus

The blue LED light (~480 nm) was used to deliver light stimuli to the retina through a ×60 water-immersion objective. The photon flux was attenuated using neutral density filters (Thor Labs). Before recording, retinas were dark-adapted for at least 5 min. The photocurrent from ipRGCs was recorded following a 50 ms full-field flash of bright light with an intensity of $6.08 \times 10^{15}$ photons · cm$^{-2}$ · s$^{-1}$.

## Electrophysiology

The ganglion cell layer of retina was visualized using IR-DIC optics at 940 nm. M4 ipRGCs, synonymous with ON-sustained alpha RGCs (*Schmidt et al., 2014*), were identified in IR-DIC as cells with large somata (>20 μm) and characteristic ON-sustained responses to increments in light, as described in *Sonoda et al., 2018*. We opted to use IR-DIC localization to initially identify the large somata of putative M4 ipRGCs because it best minimizes bleaching compared to epifluorescence. After all cellular recordings, the identity of M4 ipRGCs was confirmed by verifying the dendrites stratified only in the ON-sublamina of the inner plexiform layer (IPL) and immunolabeled with SMI-32, an M4 ipRGC marker (*Schmidt et al., 2014*; *Sonoda et al., 2018*). M1 and M2 ipRGCs were targeted in the Opn4-GFP line (M1, M2, and M3 cells labeled, M4, M5, and M6 not labeled), based on their somatic GFP signals visualized under brief epifluorescent illumination. Following recording, the identities of M1 and M2 ipRGCs were confirmed by examining dendritic stratification (M1: OFF; M2: ON) in the IPL, achieved through intracellular dye (Alexa 594) immediately following recording and confirmed post-recording by Neurobiotin fill. M2 cell dendrites stratified only in the ON-sublamina of the IPL while M1 cell dendrites stratified in the OFF-sublamina of the IPL (described in *Schmidt et al., 2008* and *Schmidt and Kofuji, 2009*). Both M1 and M2 were confirmed negative for SMI-32 immunolabeling as described in *Lee and Schmidt, 2018*. For all experiments, one cellular light response was recorded from each piece of retina to minimize light adaptation with the exception of TRPC3/6/7KO cells in mibefradil (*Figure 10*), which were stimulated twice approximately 3 min apart. Two stimulations do not affect photocurrent amplitude (data not shown).

Whole-cell recordings were performed using a Multiclamp 700B amplifier (Molecular Devices) and fire-polished borosilicate pipettes (Sutter Instruments, 3–5 MΩ for M4 cells, 5–8 MΩ for M2 and M1 cells). All voltage traces were sampled at 10 kHz, low-pass filtered at 2 kHz, and acquired using a Digidata 1550B and pClamp 10 software (Molecular Devices). All reported voltages are corrected for a –13 mV liquid junction potential calculated using Liquid Junction Potential Calculator in pClamp. We did not compensate for series resistance.

## Immunohistochemistry

After recording, retina pieces were fixed in 4% paraformaldehyde (Electron Microscopy Sciences) in 1× PBS overnight at 4°C. Retinas were then washed with 1× PBS for 3 × 30 min at room temperature (RT) and then blocked overnight at 4°C in blocking solution (2% goat serum in 0.3% Triton PBS). Retinas were then placed in primary antibody solution containing mouse anti-SMI-32 (1:1000, BioLegend, Cat# 801701, RRID:AB_509997) in blocking solution for 2–4 d at 4°C. Retinas were washed in 1× PBS for 3 × 30 min at RT and transferred to secondary antibody solution containing Alexa 488 goat anti-mouse (1:1000, Thermo, Cat# A-21131, RRID:AB_2535771) and streptavidin conjugated with Alexa 546 (1:1000, Thermo, Cat# S-11225, RRID:AB_2532130) in blocking solution overnight at 4°C. Retinas were then washed in 1× PBS for 3 × 30 min at RT and mounted using Fluoromount aqueous mounting medium (Sigma).

All images were captured using a confocal laser scanning microscope (Leica DM5500 SPE, Leica Microsystems) with a ×20 objective. To include whole dendrites of ipRGCs, tiled image stacks spanning the ganglion cell layer to inner nuclear layer were collected. Images were processed using Fiji (*Schindelin et al., 2012*).

## Slice immunohistochemistry

Mice were intracardially perfused with 4% paraformaldehyde (Electron Microscopy Sciences) in 1× PBS. The eyes were then removed and fixed in 4% paraformaldehyde (Electron Microscopy Sciences) in 1× PBS overnight at 4°C. The eyes were then washed in 1× PBS for 3 × 30 min at RT and then flash frozen in Optical Cutting Temperature Compound (Tissue Tek). The eyes were then sectioned at 20 μm on a cryostat (Leica CM1950, Leica Instruments).

The retinal slices were warmed to RT in 1× PBS for 30 min and then blocked for 2 hr at RT in blocking solution (5% donkey serum in 0.3% Triton-X in 1× PBS). Retinas were then placed in primary antibody solution containing chicken anti-mCherry (1:1000; Abcam, Cat# ab205402, RRID:AB_2722769) and rabbit anti-CaV3.1 (1:250; Alomone Labs, Cat# ACC-021, RRID:AB_2039779), rabbit anti-CaV3.2 (1:500; Alomone Labs, Cat# ACC-025, RRID:AB_2039781), or rabbit anti-CaV3.3 (1:250; Alomone Labs, Cat# ACC-009, RRID:AB_2039783) for 2 d at RT. The retinal slices were washed with 1× PBS for 5 × 15 min at RT and then incubated in secondary antibody solution containing Alexa 488 donkey anti-rabbit (1:1000; Thermo Fisher Scientific, Cat# A-21206, RRID:AB_2535792) and Alexa 594 donkey anti-Chicken (1:1000; Jackson ImmunoResearch Labs, Cat# 703-585-155, RRID:AB_2340377) for 2 hr at RT.

Images were captured using a confocal laser scanning microscope (Leica TCS SP8, Leica Microsystems) with a ×40 or ×63 objective in the Biological Imaging Facility at Northwestern (RRID:SCR_017767). Images were processed using Fiji (*Schindelin et al., 2012*).

## Data quantification and analysis

All data were analyzed using custom scripts written in MATLAB (MathWorks; RRID:SCR_001622). The scripts are available on GitHub (https://github.com/schmidtlab-northwestern/Contreras-2023-M2-Phototransductioncopy archived at *schmidtlab-northwestern, 2023*).

For voltage-clamp experiments measuring the intrinsic melanopsin response to a 50 ms light stimulus ($6.08 \times 10^{15}$ photons $\cdot$ cm$^{-2} \cdot$ s$^{-1}$), ipRGCs were voltage clamped at –66 mV, as in *Jiang et al., 2018*. Only a single cell was recorded per retina piece to ensure that the cell was not light adapted. We measured the maximum amplitude as well as the amplitude at three timepoints selected to represent different phases of the light response (highlighted in *Figure 1B*) of the photocurrent. The cells were smoothed using a 100 ms sliding average. The baseline current was quantified as the mean current in the first 3 s of the recording protocol. Maximum amplitude was calculated as the maximum current value of the smoothed trace that had the greatest change from baseline during the recording period. The Early, Intermediate, and Late timepoints represent the average change from baseline within the following timeframes after light onset: Early (141.7–440.4 ms), Intermediate (2857.7–6598.2 ms), and Late (9062.3–14062.3 ms). Values were reported as the absolute value of the current (pA), for each component.

The light-evoked I–V relationship for M2 cells was generated by recording the intrinsic melanopsin response for Control and TRPC3/6/7 KO M2 cells in response to a 50 ms light pulse. Cells were voltage clamped at potentials from –106 mV to +34 mV. Cells were allowed to stabilize at a given holding potentials for 1–3 min prior to light onset. Only a single cell was recorded per preparation to ensure dark adaptation state was as uniform as possible across cells/preparations. We plotted the I–V relationships for Maximum, Early, Intermediate, and Late amplitudes. The reversal potential was identified as where the I–V curve intersected the X-axis (i.e. I = 0).

For experiments measuring the HCN-mediated tail current, channel activation was evoked by hyperpolarizing the cell from –66 mV to –120 mV (*Van Hook and Berson, 2010*; *Chen and Yang, 2007*). Tail current amplitudes were defined as the maximum change from baseline upon return to –66 mV (after the voltage step to –120 mV) and were plotted as the absolute value of the current. Hyperpolarizing the membrane to –120 mV evoked an instantaneous current followed by the slowly activating inward current. The HCN inward current was defined as the time-dependent component at the end of the 4 s hyperpolarizing step. The HCN inward current amplitude was measured from the beginning of the slowly activating inward current (following the instantaneous current) to the end of the 4 s hyperpolarization step (*Chen and Yang, 2007*; *Van Hook and Berson, 2010*).

For experiments assessing blockade of HCN channels, cells were exposed to 50 µM ZD7288 and then subjected again to the same protocol. Blockade of HCN channels in ipRGCs was observed when the tail current amplitude was abolished (reviewed in *Biel et al., 2009*). Blockade of HCN channels, defined loss of tail currents, occurred between 5 and 8 min (5–8 min) of exposure to 50 µM ZD7288 for each individual cell. For experiments in *Figures 2A and 4B*, the HCN tail current was first measured (defined as control), and then the same cell was exposed to 50 µM ZD7288 for 5–8 min to block HCN channels. However, for *Figures 2B and 4D*, the photocurrent measurements were obtained from different cells: control and cells exposed to 50 µM ZD7288 for 5–8 min. The photocurrent was measured in different cells to avoid melanopsin photobleaching. To measure the I–V relationship of

HCN channels in M2 and M4 ipRGCs (as described in *Chen and Yang, 2007*; *Van Hook and Berson, 2010*), cells were hyperpolarizing –66 mV to –120 mV to activate the majority of HCN channels. Cells were then depolarized to various test potentials from –106 mV to –66 mV. This protocol was repeated in the same cells following 5–8 min application of 50 μM ZD7288 to ensure blockade of HCN channels. The ZD7288 sensitive tail current was derived by subtracting 5–8 min ZD7288 recordings from Control. Tail current amplitudes were measured at each test potential. The tail current amplitudes during deactivation were plotted against each test potential to construct the I–V relationship. A linear fit was used to extrapolate the reversal potential of HCN tail currents.

For experiments assessing light modulation of HCN channels, the tail current was first measured in the dark. The same cell was then exposed to light for 90 s, which allowed the inward photocurrent to stabilize. We then measured the baseline current, HCN current, and HCN tail current as described above in dark versus light. As expected, all cells in background light exhibited a steady-state inward photocurrent at –66 mV.

For experiments assessing the effects due to prolonged application of 50 μM ZD7288, cells were exposed to 20 min of ZD7288 in whole-cell patch-clamp configuration and recordings were performed following this period. Recordings for Control cells were performed 20 min after whole cell was achieved to account for effects due to long recording time. Because of the long recording times and to avoid multiple light stimulations of the same cell, which would affect photocurrent amplitude, recordings with and without ZD7288 were performed in separate cells.

Input resistance was calculated as the slope of a linear fit of the steady-state voltage deflection evoked by a series of hyperpolarizing current injections from –300 pA to 0 pA and depolarizing current injections from 0 pA to 200 pA.

Before recording calcium currents, cells were voltage clamped at –70 mV and depolarized to –40 mV to block any transient channels. Cells were then stepped from –40 mV to voltages from –60 mV to +30 mV in 10 mV intervals (*Randall and Tsien, 1997*; *Dhein et al., 2004*; *Cai et al., 2019*; *Zhang et al., 2023*).

Graphing and statistical analysis were performed using GraphPad Prism 9 software (RRID:SCR_002798). For unpaired statistical comparisons, we used a non-parametric, two-tailed Mann–Whitney $U$ test with a Bonferroni correction. For paired statistical comparisons, we used a non-parametric, two-tailed Wilcoxon matched-pairs signed-rank test. Significance was concluded when $p < 0.05$.

## Additional information

### Funding

| Funder | Grant reference number | Author |
| --- | --- | --- |
| National Eye Institute | F31EY030360 | Takuma Sonoda |
| National Eye Institute | DP2EY027983 | Tiffany M Schmidt |
| National Eye Institute | R01EY030565 | Tiffany M Schmidt |
| National Eye Institute | T32EY025202 | Jacob D Bhoi |
| National Institutes of Health | Z01-ES-101684 | Lutz Birnbaumer |
| National Heart Lung & Blood Institute | HL007909 | Ely Contreras |

The funders had no role in study design, data collection and interpretation, or the decision to submit the work for publication.

### Author contributions

Ely Contreras, Conceptualization, Investigation, Visualization, Writing - original draft, Writing – review and editing; Jacob D Bhoi, Conceptualization, Data curation, Formal analysis, Investigation, Visualization, Methodology, Writing - original draft, Writing – review and editing; Takuma Sonoda, Conceptualization, Formal analysis, Investigation, Methodology, Writing – review and editing; Lutz Birnbaumer,

Writing – review and editing, Provided reagent; Tiffany M Schmidt, Conceptualization, Supervision, Methodology, Writing - original draft, Project administration, Writing – review and editing

### Author ORCIDs
Ely Contreras ⓘ http://orcid.org/0000-0003-3684-5817
Jacob D Bhoi ⓘ http://orcid.org/0000-0002-5952-367X
Tiffany M Schmidt ⓘ http://orcid.org/0000-0002-4791-6775

### Ethics
All animals were handled according to approved institutional animal care and use committee of Northwestern University protocol IS00003845.

### Decision letter and Author response
Decision letter https://doi.org/10.7554/eLife.80749.sa1
Author response https://doi.org/10.7554/eLife.80749.sa2

---

## Additional files

### Supplementary files
• MDAR checklist

### Data availability
All data generated are included as individual points and supporting files. Source data files have been provided for all relevant figures.

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
