## [Editor Report]

Retinal ganglion cells with intrinsic photosensitivity play important emerging physiological roles. The mechanisms of phototransduction are still not well known and there exists controversy regarding the ion channels responsible for the photo response. The authors of this article present convincing data that contribute to understanding the ionic mechanisms in two of these cell types. This article will be of general interest to biologists and neuroscientists and should help resolve a major issue in retinal physiology.

---

## [Decision Letter]

**Decision letter after peer review:**

Thank you for submitting your article "Melanopsin activates divergent phototransduction pathways in ipRGC subtypes" for consideration by *eLife*. Your article has been reviewed by 3 peer reviewers, including Leon D Islas as the Reviewing Editor and Reviewer #1, and the evaluation has been overseen by Lu Chen as the Senior Editor.

Essential revisions:

1) Comparisons of current magnitudes need to be carried out after normalization to membrane capacitance to account for different cell sizes.

2) Since these cells are extensively coupled via gap junctions in a network, and have complex geometries with multiple processes, having a sense of adequate space-clamp is essential.

3) Demonstrating the functional expression, via voltage-clamp, of T-type calcium channels in M2 cells, and assessing off-target effects Mibefradil would be important steps to solidify the results, which otherwise appear preliminary.

4) It is essential to provide a good explanation for how ion channels responsible for the photo responses could be responsible for sustained, long-lasting light-evoked depolarizations.

5) Another major problem is related to the dialysis of the cellular content associated with the whole-cell patch-clamp technique, which is expected to modify intracellular signaling pathways.

*Reviewer #1 (Recommendations for the authors):*

– The title should spell in full the acronym ipRGC.

– The photocurrents from separate cells are averaged to compare between experimental groups, however, there is no indication of the expected variability due to different cell sizes. For this reason, currents should be normalized to the cell capacitance before averaging. This applies to all figures that compare photocurrents.

– Since ipRGCs are complex cells with multiple processes, it is important to have an idea of the quality of the space clamp. This information, as well as the cell input resistance and if series resistance compensation was employed, should be available in the methods section.

– The authors show evidence that M2 cells employ T-type calcium channels in their photoresponse. This evidence is based on the block of photocurrents by mibefradil, a classic T-type calcium channel blocker. However, mibefradil has been shown to also block voltage-gated and ATP-gated potassium channels and also facilitate activation of Ca++-activated potassium channels.

In other words, mibefradil is not a specific T-type Ca++ channel blocker, and just as ZD7288 might have off-target effects.

For this reason, it is important that the authors show the presence of T-type Ca++ channels in M2 cells and the absence of off-target effects of mibefradil.

*Reviewer #2 (Recommendations for the authors):*

Detailed comments:

The authors need to do a better job of explaining how HCN could mediate the melanopsin response. HCN seems an unlikely candidate as these channels would be mostly closed during the steady light-evoked depolarizations (to ~-50 mVs), and only start to activate when the ipRGCs are hyperpolarized below ~-60 mV. Thus, to me, it is not surprising that HCN doesn't play a role in mediating melanopsin responses.

While I feel it is important to directly demonstrate the negative result (in light of previous reports), I feel the results do not always support the major claims.

1) Off-target effects: The finding that ZD blocks HCN and intrinsic light responses over different time courses raises serious doubts as to whether previous pharmacological results were compromised due to off-target effects of ZD. However, the current result needs to be interpreted with caution, as the somatic depolarization used to activate HCN may only recruit channels at proximal dendrites, while light may activate HCN on the distal dendrites, deeper in the retina, which may take a longer time to block.

2) Current-voltage analysis: Distinct IV relationships for M4 photocurrents and HCN tail currents are used as an independent assay to argue against the idea that melanopsin gates HCN (Figure 3) but this is problematic for several reasons:

– IVs for HCN and light-evoked responses are expected to be different. If melanopsin did modulate HCN, currents at more negative holding potentials would be expected to be smaller compared to more depolarized potentials (see Figure 4B Jiang et al., 2018). Thus, this seems like a faulty line of reasoning.

To make the point further, the IV doesn't resemble that of T-types which are proposed to mediate the light response.

– Calling these tail currents is confusing, since tail currents are usually measured at a single holding potential (to normalize driving force), which is not what is being done here.

– Not sure why currents are reversing -100 mV (Figure 3B).

– ipRGCs are known to be coupled to amacrine cells via gap junctions, which makes the IVs difficult to interpret (since voltage-clamp can never be achieved across the network).

3) Light modulation of HCN: HCN tail currents before and after 90 sec light exposure are similar (Figure 2 S1). However, HCN tails should be measured a few seconds after the onset of the light stimulus, when photo-currents are maximum.

4) Involvement of T-types: Like HCN, T-type CaVs inactivate at depolarized potentials making it difficult to envision how they could contribute to the long-lasting photoresponses. This section is interesting but needs to be better developed.

*Reviewer #3 (Recommendations for the authors):*

This is an important work that adds to our understanding of ipRGCs phototransduction from a very reputable group in the field. From the methods, it is clear that the identification of subtypes of ipRGCs is done thoroughly and rigorously and I have no further comments on this aspect. The patch-clamp data overall seem convincing although I can't comment on this in more detail because patch-clamp recordings are not my specific area of expertise. However, while I do find the data supporting the authors' claim that ipRGCs do not signal via HCN channels which contradicts data shown by Jing et al., 2018, before invalidating the previous study I wonder whether these claims are somewhat conditions-specific. I appreciate the fact that the authors here use more physiological light levels to stimulate melanopsin, however replicating the same light stimulus (~2.5 log brighter light and a longer step 200ms) as in Jing et al., and ideally include light intensity levels in between to assess the potential involvement of HCN channels (as authors propose using ZD7288 for 5-8min incubation time) would greatly strengthen the manuscript and support the authors' claims. Also, I would suggest expanding the discussion to include authors' views on the relevance of this different phototransduction in ipRGCs subtypes and how it may impact their roles in specific visual and non-visual responses would.

---

## [Author Response]

Essential revisions:1) Comparisons of current magnitudes need to be carried out after normalization to membrane capacitance to account for different cell sizes.

We thank the reviewer for this comment and agree that any differences in capacitance between genotypes could skew interpretation of differences in current magnitudes. Both capacitance and photocurrent magnitude are subtype defining features of M1, M2, and M4 ipRGCs, and are consistent within each subtype and also differ predictably between subtypes (Schmidt and Kofuji, 2009; Ecker et al., 2010; Aranda and Schmidt, 2021). To ensure that differences in current magnitude across genotype were not due to changes in capacitance, we compared capacitance of WT versus TRPC3/6/7 KO M2 or M4 cells and found no differences. Thus, differences in capacitance do not account for differences in current magnitude across conditions, mitigating this concern. We likewise see no differences in input resistance across genotypes. These data are now included as Figure 1 —figure supplement 1 for M4 ipRGCs and Figure 5 —figure supplement 1 for M2 ipRGCs.

We have chosen to retain presentation of absolute current magnitude for light responses because the value of current magnitude is a subtype-defining feature of each ipRGC subtype, and reporting values in this format will be most informative and useful for the field when interpreting these values in the context of existing literature.

2) Since these cells are extensively coupled via gap junctions in a network, and have complex geometries with multiple processes, having a sense of adequate space-clamp is essential.

Space clamp is of course an issue with all large cells with complex geometries, and will similarly impact this study and all past studies using voltage clamp in ipRGCs (Ecker et al., 2010; Estevez et al., 2012; Jiang et al., 2018; Sonoda et al., 2018; Schmidt and Kofuji, 2009, 2010, 2011; Wong et al., 2005). In our case, we would expect space clamp be similar for a given ipRGC subtype across genotypes or pharmacological conditions, which would still allow us to interpret changes in photocurrent magnitude under different conditions. This idea is supported by the fact that neither input resistance nor capacitance of M2 or M4 ipRGCs change between the WT and TRPC3/6/7 KO genotypes (Figure 1 —figure supplement 1 and Figure 5 —figure supplement 1).

Regarding space clamp, it is also worth noting that our previous work on M4 cells, which have the largest and most complex ipRGC geometries and couple to surrounding cells, showed that the M4 photocurrent reverses right at the calculated potassium equilibrium potential (Ek) and shifts precisely as predicted when we alter external potassium concentration (Sonoda et al., 2018, Figure 7D). Additionally, the reversal of the whole cell I-V relationship precisely matches that of the M4 nucleated patch photocurrent reversal, where space clamp is not an issue (Sonoda et al., 2018, Figure 7E). This suggests that any space clamp issues are unlikely to be altering the observed reversal potential of our I-V curves. Moreover, our I-V reversals are inconsistent with currents that arise from a coupled network.

3) Demonstrating the functional expression, via voltage-clamp, of T-type calcium channels in M2 cells, and assessing off-target effects Mibefradil would be important steps to solidify the results, which otherwise appear preliminary.

We thank the reviewer for these suggestions. We have added multiple experiments to add further evidence for this interpretation.

1. Functional Expression: We now provide functional and anatomical evidence of T-type VGCC expression in M2 cells. Depolarizing voltage steps elicit inward currents in M2 cells that are reduced with the selective T-type VGCC antagonist TTA-P2, indicating that M2 ipRGCs express functional Ttype VGCCs. M1 ipRGCs, in contrast, do not exhibit similar TTA-P2 sensitive currents (Figure 9A and Figure 9 —figure supplement 2). As a second test for T-type VGCC expression, we also performed immunohistochemical labeling of retinal sections in which ipRGCs are transduced with an AAV containing a fluorescent reporter. We observed co-labeling of ON-stratifying ipRGCs with Cav3.1, 3.2, and 3.3. We include these data in Figure 9 —figure supplement 1 along with a plot of mRNA expression of these Cacna1g/h/I (corresponding to Cav 3.1/3.2/3.3) in ipRGC subtypes from a previously published single-cell RNAseq dataset (Tran et al., 2019). Collectively, these results support T-type VGCC expression in M2 ipRGCs.

2. Off Target Effects: In the first submission, we showed that the T-type antagonist Mibefradil significantly reduced the M2 photocurrent. As an additional test of the contribution of T-type VGCCs to the M2 photocurrent, we recorded the M2 photocurrent in the presence of the more selective Ttype antagonist TTA-P2 (Dreyfus et., 2010; Choe et al., 2011; Wu et al., 2018). In agreement with our findings using Mibefradil, we find that TTA-P2 nearly eliminates the M2 photocurrent. This blockade with a second, more specific T-type antagonist indicates that our findings with Mibefradil are due to blockade of T-type VGCCs and further support our initial conclusions. Importantly, M1 photocurrents were unaffected by application of TTA-P2 alone and were also unaffected by application of a cocktail of VGCC blockers containing Mibefradil, arguing against non-specific effects of these antagonists on TRPC channels (Figures 8,9). T-type blockade also had no effect on the magnitude of the HCN tail current (Figure 9 —figure supplement 3).

We include these new results as main figures in the paper and have re-ordered this section of the paper for clarity and flow.

4) It is essential to provide a good explanation for how ion channels responsible for the photo responses could be responsible for sustained, long-lasting light-evoked depolarizations.

We appreciate this comment and have added in the discussion of a potential mechanism through which T-type VGCCs could mediate the sustained current observed in M2s in Lines 440-453.

5) Another major problem is related to the dialysis of the cellular content associated with the whole-cell patch-clamp technique, which is expected to modify intracellular signaling pathways.

We agree that perforated patch has the advantage of preventing dialysis of intracellular components. However, a major goal of this work is to reconcile discrepancies in models from the Jiang and Sonoda studies, which both used whole-cell patch clamp recordings (Sonoda et al., 2018; Jiang et al., 2018). Because of this need for comparison we have chosen to prioritize methodological consistency with past work in our experimental design.

Reviewer #1 (Recommendations for the authors):– The title should spell in full the acronym ipRGC.

In the title, “ipRGC” was replaced with “intrinsically photosensitive retinal ganglion cell.”

– The photocurrents from separate cells are averaged to compare between experimental groups, however, there is no indication of the expected variability due to different cell sizes. For this reason, currents should be normalized to the cell capacitance before averaging. This applies to all figures that compare photocurrents.

We address this point in response to Essential revisions point 1 above.

– Since ipRGCs are complex cells with multiple processes, it is important to have an idea of the quality of the space clamp. This information, as well as the cell input resistance and if series resistance compensation was employed, should be available in the methods section.

Series resistance was not compensated for, and this is now more clearly stated in the methods. We discuss space clamp in our response to Essential revisions point 2 above.

– The authors show evidence that M2 cells employ T-type calcium channels in their photoresponse. This evidence is based on the block of photocurrents by mibefradil, a classic T-type calcium channel blocker. However, mibefradil has been shown to also block voltage-gated and ATP-gated potassium channels and also facilitate activation of Ca++-activated potassium channels.In other words, mibefradil is not a specific T-type Ca++ channel blocker, and just as ZD7288 might have off-target effects.For this reason, it is important that the authors show the presence of T-type Ca++ channels in M2 cells and the absence of off-target effects of mibefradil.

We thank the reviewer for this important point. These points are largely addressed in Essential revisions point 3 above. Briefly, we now include data from a second, more specific, T-type antagonist, TTA-P2, which nearly eliminates the M2 photocurrent but does not affect the M1 photocurrent (Figure 9), mirroring our results using Mibefradil (now shown in Figure 10). Figure 9 —figure supplement 1 now plots mRNA expression of Cacna1g/h/i (i.e. Cav 3.1/3.2/3.3) across ipRGC subtypes and shows immunohistochemical labeling for these subunits in retinal sections where ipRGCs are labeled with a fluorescent reporter and Figure 9 —figure supplement 2 shows functional evidence of T-type VGCCs in M2 cells. All of our results from these newly added experiments support expression of functional T-type VGCCs in M2 cells and a key role for these channels in M2 phototransduction, further strengthening our initial conclusions.

Reviewer #2 (Recommendations for the authors):Detailed comments:The authors need to do a better job of explaining how HCN could mediate the melanopsin response. HCN seems an unlikely candidate as these channels would be mostly closed during the steady light-evoked depolarizations (to ~-50 mVs), and only start to activate when the ipRGCs are hyperpolarized below ~-60 mV. Thus, to me, it is not surprising that HCN doesn't play a role in mediating melanopsin responses.While I feel it is important to directly demonstrate the negative result (in light of previous reports), I feel the results do not always support the major claims.1) Off-target effects: The finding that ZD blocks HCN and intrinsic light responses over different time courses raises serious doubts as to whether previous pharmacological results were compromised due to off-target effects of ZD. However, the current result needs to be interpreted with caution, as the somatic depolarization used to activate HCN may only recruit channels at proximal dendrites, while light may activate HCN on the distal dendrites, deeper in the retina, which may take a longer time to block.

We thank the reviewer for this point. If penetration issues were to blame, we would expect to see at least partial reduction of the M2 and M4 photocurrents in 5-8 minutes of ZD7288. However, despite blockade of all detectable HCN current evoked by somatic hyperpolarization under these conditions, the photocurrent of M2 and M4 cells was completely unchanged. Additionally, the dendrites of both M4 and M2 ipRGCs stratify extremely close to RGC cell bodies, in sublamina 5 of the IPL. We would therefore expect similar drug penetration for the dendrites versus soma.

The major goal of the 20 minute versus 5-8 minute incubation of ZD7288 was to demonstrate that we are able to fully replicate previous results and provide a plausible explanation for their initial interpretation that HCN channels play a role in M2 and M4 phototransduction. On the other hand, T-type VGCC antagonists have no effect on M2 HCN tail current amplitude but nearly eliminates the in M2 photocurrent.

Importantly, we are not making these interpretations in isolation, but are considering them alongside multiple, converging lines of evidence against HCN involvement in M4 phototransduction including (1) the I-V slope and reversal potential of the M4 photocurrent at Ek (in whole cell and nucleated patch configurations and using multiple durations of light stimulation, See Sonoda et al. 2018 Figures 7C-D, Figure 7E-F, and Figure S6) (2) increased input resistance and excitability of M4 cells in light (See Figure 8 from Sonoda et al. 2018) and (3) blockade of the M4 photocurrent with a 2-pore potassium channel antagonist (Figure S7 from Sonoda et al., 2018), all of which are consistent with closure of potassium channels as the M4 transduction channel. For M2 cells, in addition to the results using ZD7288, we find that (1) the M2 photocurrent reverses at +19 mV, which is distinct from the extrapolated reversal of the M2 HCN tail current at -32 mV and (2) the M2 photocurrent is reduced by ~75% by application of the Ttype VGCC antagonist TTA-P2, application of the T-type VGCC antagonist Mibefradil, application of a cocktail of VGCC blockers, and by genetic knockout of TRPC3/6/7.

2) Current-voltage analysis: Distinct IV relationships for M4 photocurrents and HCN tail currents are used as an independent assay to argue against the idea that melanopsin gates HCN (Figure 3) but this is problematic for several reasons:– IVs for HCN and light-evoked responses are expected to be different. If melanopsin did modulate HCN, currents at more negative holding potentials would be expected to be smaller compared to more depolarized potentials (see Figure 4B Jiang et al., 2018). Thus, this seems like a faulty line of reasoning.To make the point further, the IV doesn't resemble that of T-types which are proposed to mediate the light response.– Calling these tail currents is confusing, since tail currents are usually measured at a single holding potential (to normalize driving force), which is not what is being done here.– Not sure why currents are reversing -100 mV (Figure 3B).– ipRGCs are known to be coupled to amacrine cells via gap junctions, which makes the IVs difficult to interpret (since voltage-clamp can never be achieved across the network).

We thank the reviewer for these important points. We address each below.

Regarding HCN tail current I-V: We have made our language more precise around our intent to compare the reversal potential and permeation properties of M2 and M4 HCN channels versus photocurrent, and have removed discussion of the slope and magnitude of the tail current versus photocurrent. If melanopsin phototransduction opens HCN channels, the reversal potential of the photocurrents and extrapolated reversal potential of the HCN tail currents should align for each cell type because the reversal potential is a reflection of the permeation properties of a given channel(s). This was the main purpose of our including this analysis of the extrapolated reversal (as performed in Van Hook and Berson, 2010). In fact, the HCN tail current reversal potential does not align with that of the photocurrent either M4 or M2 ipRGCs, and we have edited figures and wording throughout the paper to focus on this point. Importantly, HCN currents would not, in practice, reverse at all, and the photocurrents of both M2 and M4 ipRGCs do in fact reverse. Importantly, these differences in reversal potential are considered alongside other pieces of supporting evidence. For example: (1) The M4 photocurrent is not blocked by ZD7288 (2) The photocurrent in TRPC3/6/7 KO M4 cells (where HCN would be the only contributor if it were the transduction channel) continues to show increasing inward current at positive potentials well beyond where HCN channels would be expected to conduct inward current and (3) The M4 photocurrent reverses at the calculated Ek, and has a detectable outward current below -90mV, which are both inconsistent with HCN conductance.

Regarding point 3 about reversal of M4 photocurrent at -100 mV: This data point was inadvertently left out when we replotted the original data from Sonoda et al., 2018, and we now include it in the graph. We apologize for the omission.

Regarding the shape of the M2 photocurrent I-V relationship compared to that of T-type VGCCs: Our evidence in M2 cells clearly demonstrate that the photocurrent is a mixed conductance, and so we would not expect the shape of the I-V relationship to match one type of channel. Importantly, we now provide functional evidence in Figure 9 —figure supplement 1 and 2 that M2 cells functionally express T-type VGCCs and have added additional pharmacological evidence in support of T-type VGCCs as an M2 phototransduction channel.

Regarding coupling: We have previously shown that the reversal potential of the M4 photocurrent in whole cell conditions falls at the calculated Ek and is identical to the reversal potential of the M4 photocurrent in a nucleated patch configuration where there is no coupling and the membrane can be most adequately voltage clamped (Sonoda et al., 2018, Figure 7). This argues against coupling or space clamp having a major impact on our M4 cell I-V results or interpretation.

3) Light modulation of HCN: HCN tail currents before and after 90 sec light exposure are similar (Figure 2 S1). However, HCN tails should be measured a few seconds after the onset of the light stimulus, when photo-currents are maximum.

We thank the reviewer for this point. If there is a persistent photocurrent after 90 seconds of background light exposure, then it will be possible to assess whether that current is indeed through HCN channels. The ipRGC light response can persist for minutes to hours in all ipRGCs (Berson et al., 2002; Hattar et al., 2002; Wong et al., 2005; Wong et al., 2012; Emanuel and Do 2015). Indeed, we see that the amplitude of the holding current for M2 and M4 ipRGCs in these experiments at our baseline holding potential of -66 mV is more negative in all cells following 90 seconds of light exposure, indicating that the transduction channels in M4 and M2 ipRGCs are conducting inward current during this period of light exposure. We now include a quantification of this as new panels in Figure 2 —figure supplement 1 and Figure 4 —figure supplement 1.

We chose to make these measurements after the light response had reached a steady state because it would be difficult to ensure we are consistently capturing the maximum of the photocurrent for individual cells and to then apply the relatively long HCN voltage clamp protocol in that timeframe in a consistent way. This would lead to unpredictable variability and make it difficult to interpret our results. Given the difficulty of standardizing for when this expected maximum would occur, and because the photocurrent is highly dynamic in the initial stages (Figure 1 and 4), it would be difficult to compare and interpret differences in magnitude if the voltage protocol were applied earlier in the light response.

4) Involvement of T-types: Like HCN, T-type CaVs inactivate at depolarized potentials making it difficult to envision how they could contribute to the long-lasting photoresponses. This section is interesting but needs to be better developed.

This is an important point, and we thank the reviewer for requesting that we expand upon it. We have added in the discussion of a potential mechanism through which T-type VGCCs could mediate the sustained current observed in M2s in Lines 440-453.

Reviewer #3 (Recommendations for the authors):This is an important work that adds to our understanding of ipRGCs phototransduction from a very reputable group in the field. From the methods, it is clear that the identification of subtypes of ipRGCs is done thoroughly and rigorously and I have no further comments on this aspect. The patch-clamp data overall seem convincing although I can't comment on this in more detail because patch-clamp recordings are not my specific area of expertise. However, while I do find the data supporting the authors' claim that ipRGCs do not signal via HCN channels which contradicts data shown by Jing et al., 2018, before invalidating the previous study I wonder whether these claims are somewhat conditions-specific. I appreciate the fact that the authors here use more physiological light levels to stimulate melanopsin, however replicating the same light stimulus (~2.5 log brighter light and a longer step 200ms) as in Jing et al., and ideally include light intensity levels in between to assess the potential involvement of HCN channels (as authors propose using ZD7288 for 5-8min incubation time) would greatly strengthen the manuscript and support the authors' claims.

It is important to note that the intensity used here is saturating and produces a maximum photocurrent, making it unlikely that there would be no detectable contribution of HCN channels if the melanopsin cascade is already maximally activated. We do appreciate this comment and agree that our study cannot fully preclude the possibility that HCN channels are activated by the phototransduction cascade at brighter light intensities. Our equipment is not able to produce a light stimulus that is brighter than that used in this study. We have added an acknowledge of this limitation in the discussion.

Also, I would suggest expanding the discussion to include authors' views on the relevance of this different phototransduction in ipRGCs subtypes and how it may impact their roles in specific visual and non-visual responses would.

We have expanded the discussion of this in the paper. Notably, the behaviors to which M2 cells contribute have not yet been identified and are an important avenue for future research in the field.

References

Aranda, M. L., and Schmidt, T. M. (2021). Diversity of intrinsically photosensitive retinal ganglion cells: circuits and functions. Cellular and molecular life sciences: CMLS, 78(3), 889–907.

Berson, D. M., Dunn, F. A., and Takao, M. (2002). Phototransduction by retinal ganglion cells that set the circadian clock. Science (New York, N.Y.), 295(5557), 1070–1073.

Biel, M., Wahl-Schott, C., Michalakis, S., and Zong, X. (2009). Hyperpolarization-activated cation channels: from genes to function. Physiological reviews, 89(3), 847–885.

Chen, L., and Yang, X. L. (2007). Hyperpolarization-activated cation current is involved in modulation of the excitability of rat retinal ganglion cells by dopamine. Neuroscience, 150(2), 299–308.

Choe, W., Messinger, R. B., Leach, E., Eckle, V. S., Obradovic, A., Salajegheh, R., JevtovicTodorovic, V., and Todorovic, S. M. (2011). TTA-P2 is a potent and selective blocker of T-type calcium channels in rat sensory neurons and a novel antinociceptive agent. Molecular pharmacology, 80(5), 900–910.

Ecker, J. L., Dumitrescu, O. N., Wong, K. Y., Alam, N. M., Chen, S.-K., LeGates, T., Renna, J. M., Prusky, G. T., Berson, D. M., and Hattar, S. (2010). Melanopsin-expressing retinal ganglioncell photoreceptors: cellular diversity and role in pattern vision. Neuron 67, 49-60.

Emanuel, A. J., and Do, M. T. (2015). Melanopsin tristability for sustained and broadband phototransduction. *Neuron*, *85*(5), 1043–1055.

Estevez, M. E., Fogerson, P. M., Ilardi, M. C., Borghuis, B. G., Chan, E., Weng, S., Auferkorte, O. N., Demb, J. B., and Berson, D. M. (2012). Form and function of the m4 cell, an intrinsically photosensitive retinal ganglion cell type contributing to geniculocortical vision. J. Neurosci. 32, 13608-13620.

Dreyfus, F. M., Tscherter, A., Errington, A. C., Renger, J. J., Shin, H. S., Uebele, V. N., Crunelli, V., Lambert, R. C., and Leresche, N. (2010). Selective T-type calcium channel block in thalamic neurons reveals channel redundancy and physiological impact of I(T)window. The Journal of neuroscience: the official journal of the Society for Neuroscience, 30(1), 99–109.

Hattar, S., Liao, H. W., Takao, M., Berson, D. M. and Yau, K. W. (2002). Melanopsin-containing retinal ganglion cells: architecture, projections, and intrinsic photosensitivity. Science 295, 10651070.

Jiang, Z., Yue, W., Chen, L., Sheng, Y., and Yau, K. W. (2018). Cyclic-Nucleotide- and HCNChannel-Mediated Phototransduction in Intrinsically Photosensitive Retinal Ganglion Cells. Cell, 175(3), 652-664.e12.

Schmidt, T. M., and Kofuji, P. (2009). Functional and morphological differences among intrinsically photosensitive retinal ganglion cells. The Journal of neuroscience: the official journal of the Society for Neuroscience, 29(2), 476-482.

Schmidt, T. M., and Kofuji, P. (2011). Structure and function of bistratified intrinsically photosensitive retinal ganglion cells in the mouse. The Journal of comparative neurology, 519(8), 1492–1504.

Schmidt, T. M., Taniguchi, K., and Kofuji, P. (2008). Intrinsic and extrinsic light responses in melanopsin-expressing ganglion cells during mouse development. Journal of neurophysiology, 100(1), 371–384.

Sonoda, T., Lee, S. K., Birnbaumer, L., and Schmidt, T. M. (2018). Melanopsin Phototransduction Is Repurposed by ipRGC Subtypes to Shape the Function of Distinct Visual Circuits. Neuron, 99(4), 754–767.e4.

Tran, N. M., Shekhar, K., Whitney, I. E., Jacobi, A., Benhar, I., Hong, G., Yan, W., Adiconis, X., Arnold, M. E., Lee, J. M., Levin, J. Z., Lin, D., Wang, C., Lieber, C. M., Regev, A., He, Z., and Sanes, J. R. (2019). Single-Cell Profiles of Retinal Ganglion Cells Differing in Resilience to Injury Reveal Neuroprotective Genes. Neuron, 104(6), 1039–1055.e12.

Van Hook, M. J., and Berson, D. M. (2010). Hyperpolarization-activated current (I(h)) in ganglioncell photoreceptors. PloS one, 5(12), e15344.

Wong K. Y. (2012). A retinal ganglion cell that can signal irradiance continuously for 10 hours. The Journal of neuroscience: the official journal of the Society for Neuroscience, 32(33), 11478–11485.

Wong, K. Y., Dunn, F. A., and Berson, D. M. (2005). Photoreceptor adaptation in intrinsically photosensitive retinal ganglion cells. Neuron 48, 1001-1010.

Wu, J., Peng, S., Xiao, L., Cheng, X., Kuang, H., Zhu, M., Zhang, D., Jiang, C., and Liu, T. (2018). Cell-Type Specific Distribution of T-Type Calcium Currents in Lamina II Neurons of the Rat Spinal Cord. Frontiers in cellular neuroscience, 12, 370.